# The first mitotic division of human embryos is highly error prone

Cerys E. Currie[1,2,7], Emma Ford[1,2,7], Lucy Benham Whyte[3,6], Deborah M. Taylor[3], Bettina P. Mihalas[4], Muriel Erent[1,2], Adele L. Marston [4], Geraldine M. Hartshorne [1,3,5] ✉ & Andrew D. McAinsh [1,2,5] ✉

Human beings are made of ~50 trillion cells which arise from serial mitotic divisions of a single cell - the fertilised egg. Remarkably, the early human embryo is often chromosomally abnormal, and many are mosaic, with the karyotype differing from one cell to another. Mosaicism presumably arises from chromosome segregation errors during the early mitotic divisions, although these events have never been visualised in living human embryos. Here, we establish live cell imaging of chromosome segregation using normally fertilised embryos from an egg-share-to-research programme, as well as embryos deselected during fertility treatment. We reveal that the first mitotic division has an extended prometaphase/metaphase and exhibits phenotypes that can cause nondisjunction. These included multipolar chromosome segregations and lagging chromosomes that lead to formation of micronuclei. Analysis of nuclear number and size provides evidence of equivalent phenotypes in 2-cell human embryos that gave rise to live births. Together this shows that errors in the first mitotic division can be tolerated in human embryos and uncovers cell biological events that contribute to preimplantation mosaicism.

Human reproduction is remarkably inefficient, with only ~30% of all conceptions resulting in live birth[1]. A frequent feature of early human pregnancy loss is the presence of aneuploidies, in which cells of the embryo or foetus contain an incorrect set and/or structural rearrangement of chromosomes[2,3]. Two patterns of aneuploidy can be recognised in human embryos: (1) Homogeneous aneuploidies where all cells of an embryo carry the same, incorrect chromosome complement. These must originate from chromosome segregation errors during gamete formation, most commonly the oocyte's meiotic divisions[4–6]. Such errors involve nondisjunction of bivalents and premature separation of sister chromatids, arising from improper chromosome-spindle attachments and/or age-dependent weakening of cohesion[5,7–9]. Gain of an extra chromosome 13, 18 or 21 in all cells of an embryo is compatible with live birth but results in neuro-developmental conditions, while sex chromosome aneuploidies may result in subfertility. Aneuploidies of the remaining chromosomes are lethal[10]. Such events can help explain the low fertility in teenage girls and in women of advancing maternal age[11]. Sperm cells can also be aneuploid, but at significantly lower rates[12]. (2) Mosaic aneuploidy where embryos comprise mixtures of euploid and aneuploid cell lineages that arise after fertilisation[13]. Importantly, this aneuploidy is not age-related, and is identified frequently by genetic screening of human preimplantation embryos[14]. Studies describing single cell analysis of separated blastomeres showed that a high fraction (>75%) of day 3–6 (8-cell onwards) human embryos possess a mosaic aneuploid phenotype[15,16]. These embryos were created through assisted reproduction, however levels are hypothesised to be similar in the general population[13]. Remarkably, healthy babies with correct chromosome

[1]Division of Biomedical Sciences, Warwick Medical School, University of Warwick, Coventry CV4 7AL, UK. [2]Centre for Mechanochemical Cell Biology, University of Warwick, Coventry CV4 7AL, UK. [3]University Hospitals Coventry and Warwickshire NHS Trust, Coventry CV2 2DX, UK. [4]Wellcome Centre for Cell Biology, University of Edinburgh, Edinburgh, UK. [5]Centre for Early Life, University of Warwick, Coventry CV4 7AL, UK. [6]Present address: Kings Fertility Ltd, Fetal Medicine Research Institute, 16-20 Windsor Walk, SE5 8SS London, UK. [7]These authors contributed equally: Cerys E. Currie, Emma Ford. ✉e-mail: geraldine.hartshorne@warwick.ac.uk; a.d.mcainsh@warwick.ac.uk

numbers can be born from mosaic embryos[17–19], however severe mosaic aneuploidies are associated with arrest before the blastocyst stage, implantation failure and pregnancy loss[19–21]. Moreover, as aneuploidy originating after fertilisation is not age-dependent, embryos from women under age 40 years can often be mosaic[14]. The transfer of known mosaic embryos during assisted reproduction treatment is therefore debated[22].

Mosaic aneuploidy is presumed to arise from chromosome segregation errors during the first few mitotic cleavage divisions of the preimplantation embryo[13]. Evidence for this comes from observation of spindle and nuclear abnormalities in day 3 and 5 fixed human embryos[23,24], and indirectly from sequencing data[14,16,25–27]. Furthermore, mitotic segregation errors have been directly observed by live chromosome imaging of bovine and mouse embryos during the first cleavage division[28–30]. Mouse embryos rarely mis-segregate chromosomes during the first division (~2%)[30], while bovine cleavage stage embryos display levels of aneuploidy comparable with human (~70%), suggesting a much higher error rate[31]. In mouse, embryos displaying chromosome errors and micronuclei arising in early cleavage divisions (1 to 8-cell) are able to retain their developmental potential, however errors can be associated with reduced blastocyst formation[29]. These data suggest that the early mitoses of the human embryo may have an error prone phenotype that causes mosaic aneuploidy.

Exactly how and when mitotic errors occur in human preimplantation embryos remains unknown, largely because chromosome segregation has, to our knowledge, never been visualised live in this context. Chromosome movements cannot be seen in brightfield microscopy used in routine clinical embryology but requires fluorescent labelling and live imaging. Here, we present live-embryo chromosome imaging methods using high quality, non-thawed, human zygotes. This allows us to directly observe chromosome movements throughout the first two divisions in live human embryos and characterise the timing of mitotic events and associated phenotypes. Our experiments provide the initial insight into timing and fidelity of chromosome segregation at the start of human life and sheds light on the origins of mosaicism in human embryos.

## Results

### High-quality human embryos display phenotypes associated with frequent chromosome mis-segregation

To access high-quality, fresh human zygotes, we established an 'egg-share-to-research' programme where four young women donated half of their collected oocytes to research with the other half used for their own treatment during an in vitro fertilisation (IVF) cycle (see 'Methods' for details). Following random allocation, the donated mature oocytes (metaphase II) were fertilised via intracytoplasmic sperm injection (ICSI) using sperm from the same donor (Supplementary Fig. 1a). Each 'egg-share' embryo was therefore known to be correctly fertilised with a single sperm resulting in a 2 pronuclear (PN) zygote. To investigate the origins of post-fertilisation, mitotic aneuploidy we carried out time-lapse imaging of research embryos. The glycoprotein-rich zona pellucida was removed and SiR-DNA, a far-red fluorogenic probe for DNA, added to visualise chromosome dynamics (Fig. 1a and Supplementary Movie 1)[32]. In parallel, patient treatment embryos were monitored in the clinic via time-lapse imaging with Hoffman contrast (EmbryoScope™) to inform embryo selection for transfer to the patient (Fig. 1b). This allowed us to compare autologous timings between our live-embryo chromosome imaging movies and clinical embryos used in that patient's treatment (Fig. 1c).

Initial analysis showed that the duration between pronuclear breakdown (PNBD), furrow ingression and completion of cytokinesis (2-cell stage) was consistent between treatment and research embryos where the full first division was recorded (Fig. 1c and Supplementary Table 1), suggesting that the fluorescent imaging methodology did not impair embryo progression. Live-embryo chromosome imaging allows

us to determine the timing of key chromosome-related events during the first mitosis that are invisible in clinical image sequences. Following pronuclear breakdown ($T = 0$ min), embryos took ~135 min (IQR = 15 min) to align chromosomes into a metaphase plate and initiate segregation of sister chromatids (Fig. 1d and Supplementary Fig. 1b). These timings are substantially longer than those of human somatic cells (~24 min[33]). Similarly, the time from anaphase onset to the initiation of furrow ingression was ~45 min, compared to ~7 min in somatic cells[34]. Taking all phases together, the total median duration of the first human embryonic mitosis was 195 mins (Fig. 1d and Supplementary Fig. 1b).

We also observed that 33% of the 12 egg-share embryos proceeded through the first anaphase with phenotypes that are associated with chromosome segregation errors (Fig. 1e, f). These consisted of lagging chromosomes–chromosomes that dwell at the spindle equator while others move to opposite spindle poles (Fig. 1e, purple boxes, Supplementary Movies 2 and 3), and one case of multipolar chromosome segregation in which chromosomes are separated at anaphase into more than two masses (Fig. 1e, green box). Lagging chromosomes can lead to nondisjunction and are indicative of merotelic attachments that cause chromosomes to mis-segregate[35] while multipolar segregation most likely leads to aneuploid daughter cells due to the unequal division[36]. These initial data indicate that the first embryonic mitosis in humans is more error-prone than human somatic cells, where the frequency of lagging chromosomes is relatively rare (1–6%)[37,38]. This phenotype points to a potential origin of mosaicism that is well documented in early human preimplantation embryos (see 'Introduction').

### Deselected human embryos are also error prone and show evidence of micronuclei formation

Inevitably, the sample size is limited given the small number of patients on such an egg-share-to-research programme. We therefore made use of a second, more abundant source of 41 deselected embryos donated by consenting patients during fertility treatment (Supplementary Table 2). Deselected embryos consist of mis-fertilised zygotes that have a number of pronuclei varying from the expected 2PN: mono-pronuclear (1PN) or tripronuclear (3PN), and embryos which appeared unfertilised at the time of assessment (delayed fertilisation, 0PN). This material is unsuitable for patient treatment and would otherwise be discarded. Of the 33 deselected embryos which were imaged during anaphase, 51.5% ($n = 17$) underwent bipolar chromosome segregation with 23.5% ($n = 4$) of those displaying a lagging chromosome during anaphase (Fig. 2a, b second bar). This is consistent with egg-share-to-research embryos, where 30% of bipolar segregations displayed lagging chromosomes (3/10, Fig. 2b first bar, $p = 1$). Pooling all filmed embryos together gives an overall incidence of those with lagging chromosomes of 25.9% (7/27). Importantly, in 14.8% of embryos displaying bipolar segregation, lagging chromosomes developed into clear micronuclei during imaging (4/27, Fig. 2b fourth bar, Fig. 2d and Supplementary Movie 4). Micronuclei are well established to be associated with the development of aneuploidy in somatic cells[39]. This includes both numerical aneuploidies and segmental aneuploidies, the latter of which is caused by chromothripsis and leads to an elevated mutation rate[40]. However, whether the same mechanisms are present in human embryos remains unknown.

Multipolar chromosome segregation was more frequent in the deselected embryos dataset (48.4%, Fig. 2a, c second bar), and occasionally two metaphase plates appeared to overlap perpendicularly, or even form separately in the zygote (Supplementary Fig. 2a, b and Supplementary Movies 5 and 6). This result can be partially explained by a pronuclei number other than two. Where pronuclear status was known (n = 10), 60% of multipolar divisions occurred in tripronuclear (3PN) embryos from IVF (Fig. 2c, third bar), which could arise due to the presence of supernumerary

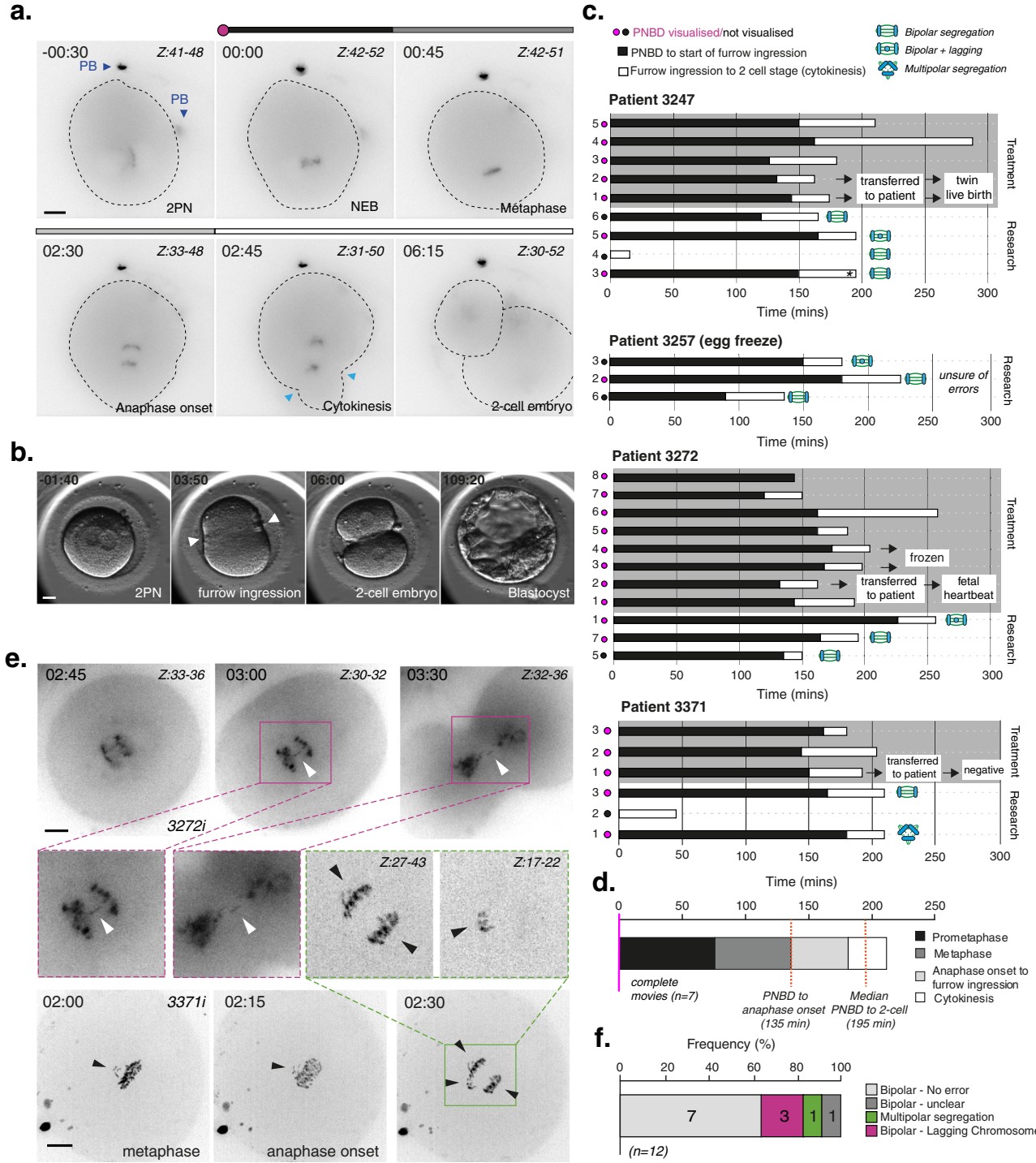

**Fig. 1 | Clinical-grade human embryos display high levels of chromosome segregation errors during the first mitotic division. a** Time lapse imaging of a representative egg-share-to-research human embryo (3247iii) progressing through the first embryonic mitosis. Chromosomes are visualised using SiR-DNA dye. Z indicates slices shown as a maximum intensity projection. Time in hours:mins, scale bar 20 μm. Light blue arrows indicate onset of cleavage furrow ingression, PB = polar body. **b** Representative movie stills showing clinical EmbryoScope monitoring of an egg-share-to-research embryo used for patient treatment. White arrows denote onset of cleavage furrow ingression. Scale bar: 20 μm, time in hours:mins. **c** History plots of egg-share embryos; half used for patient treatment (grey background) and half donated to research (white background). Black and white bars denote timings of critical stages during the first embryonic mitosis. Pink/black dots indicate whether pronuclear break down (PNBD) was visualised during filming.

Asterisk denotes images of embryo shown in (**a**). The research embryos of patients 3247, 3273 and 3257 were imaged using a widefield microscope, the research embryos of patient 3371 were imaged using a spinning disk microscope. Embryos with a black dot started dividing before imaging was started. **d** Median durations of each mitotic phase plotted consecutively from 7 complete egg-share embryos movies during the first mitosis. Red lines indicate median times for the population. **e** Time lapse imaging of an egg-share-to-research embryo undergoing the first embryonic mitosis in the presence of anaphase lagging chromosomes (purple boxes, white arrows) (3272i), and a multipolar division (3371i) (green boxes, black arrows). Chromosomes are visualised using SiR-DNA dye. Z indicates slices shown as a maximum intensity projection. Time in hours:mins, scale bar 20 μm.
**f** Quantification of anaphase errors in egg-share-to-research embryos visualised by chromosome imaging. Source data are provided as a Source data file.

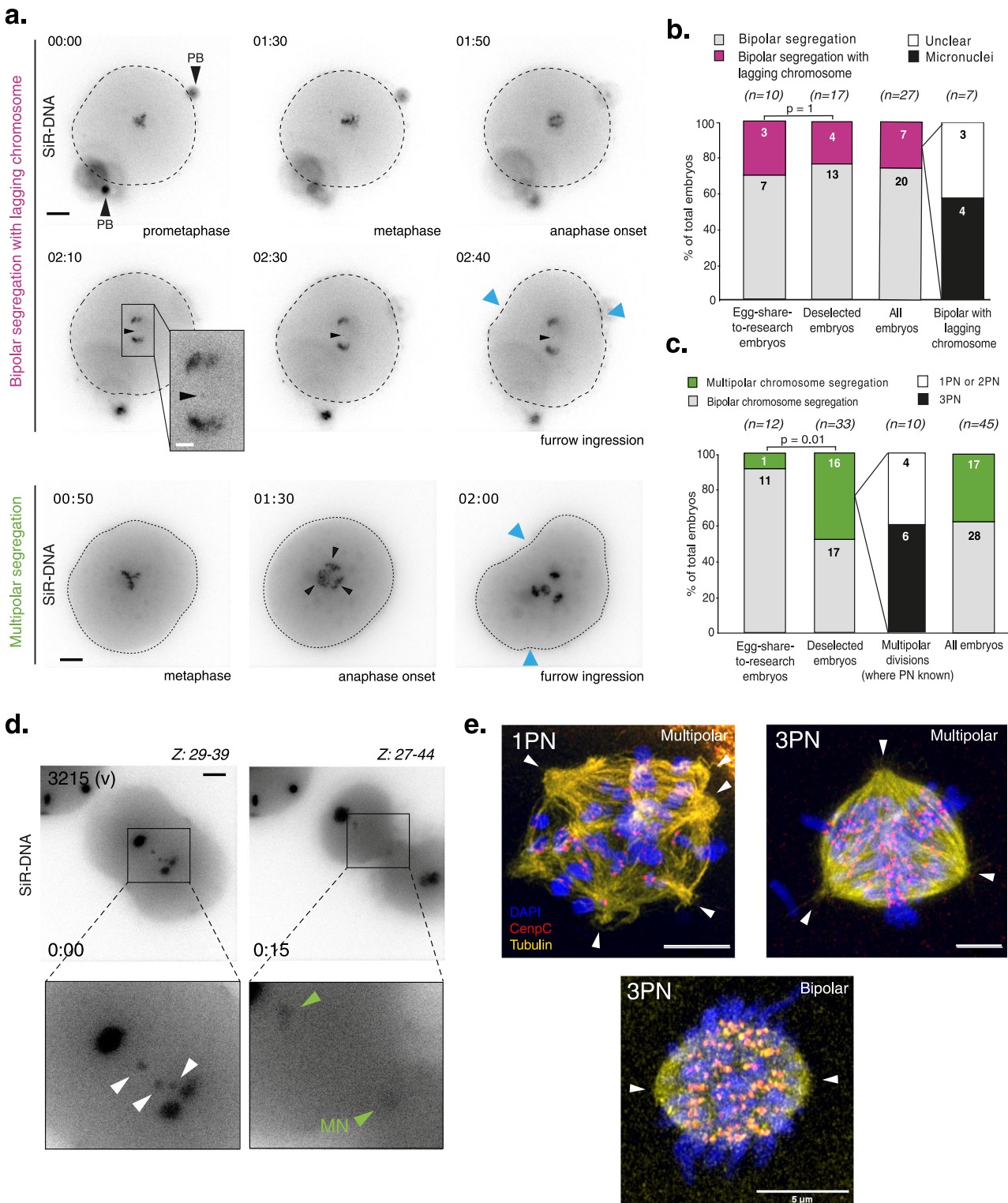

centrosomes brought by two individual sperm, generating abnormal mitotic spindles[41]. However, 40% of embryos dividing in a multipolar fashion where PN status was known ($n = 10$) were either bipronuclear or monopronuclear. Moreover, using super resolution microscopy we directly observed multipolar mitotic spindles in both 3PN (as expected) and 1PN human embryos (Fig. 2e) fixed during the first mitosis. This suggests that multipolar chromosome segregation is possible in the absence of supernumerary centrosomes and confirms the observation made in egg-sharer-to-research embryos (Fig. 1e, f).

To rule out any effects from SiR-DNA and/or removal of the zona pellucida on the error rate, we injected mRNA encoding Histone2B-mCherry into a small cohort of delayed fertilised (0PN) embryos deselected from IVF treatment. Using spinning disk confocal live cell imaging (Fig. 3a, b and Supplementary Movie 7) we observed similar error prone phenotypes, with 2/3 embryos displaying clear errors in the form of anaphase lagging chromosomes (Fig. 3c and Supplementary Movie 8) and multipolar segregation (Fig. 3d and Supplementary Movie 9). One embryo formed clear micronuclei (Fig. 3c), while another divided with a lagging

**Fig. 2 | The first mitosis in deselected human embryos is highly error prone, consistent with clinical-grade embryos. a** Top panel: Time lapse imaging of a deselected human embryo progressing through the first embryonic mitosis with a lagging chromosome (Embryo 3004iii). Bottom panel: Time lapse imaging of a deselected human embryo of unknown pronuclei status progressing through the first embryonic mitosis with multipolar chromosome segregation (Embryo 3034vii). Chromosomes are visualised using SiR-DNA dye. Z indicates slices shown as a maximum intensity projection. Time in hours:mins, scale bar 20 μm. Blue arrows indicate onset of cleavage furrow ingression. Black arrows indicate polar bodies. All deselected embryos were imaged using a widefield microscope. **b** Quantification of embryos undergoing the first embryonic mitosis with bipolar chromosome segregation. N numbers are shown within bars. The number of embryos in which micronuclei clearly formed around lagging chromosomes are shown in the fourth bar. P value from a two-sided Fisher's exact test.

**c** Quantification of embryos undergoing bipolar or multipolar divisions in the first embryonic mitosis. N numbers are shown within bars. As deselected embryos can have varying numbers of pronuclei, this was detailed for embryos dividing with multipolar chromosome segregation (third bar). All egg-share-to-research embryos contain 2 pronuclei. P value from a two-sided Fishers exact test. **d** Time lapse imaging of a deselected human embryo progressing through the first embryonic mitosis in the presence of lagging chromosomes (white arrows), around which micronuclei form (green arrows). Chromosomes are visualised using SiR-DNA dye. Z indicates slices shown as a maximum intensity projection, time in hours:mins, scale bar 20 μm. (Embryo 3215 v). **e** Airyscan super-resolution confocal microscopy images of 1PN and 3PN human embryos fixed during the first mitotic division and stained with DAPI, CenpC and tubulin antibodies. White arrows indicate perceived MTOCS/spindle poles. Scale bar 5 μm. Source data are provided as a Source data file.

chromosome (Fig. 3b) but did not generate a micronucleus at the 2-cell stage.

## Mitotic timing is comparable in embryos used in patient treatment

We further investigated the timing of mitotic events in deselected embryos imaged using SiR-DNA where PNBD was observed (n = 18) and found that duration of prometaphase, metaphase and cytokinesis were similar to the egg-sharer embryos (Fig. 4a). Plotting metaphase time versus prometaphase time (all embryos) reveals an inverse relationship suggesting that a timing mechanism may operate to fix the duration of PNBD to anaphase onset at ~135 min (Supplementary Fig. 3b). Combining research and egg-sharer embryos together (n = 24) gives an overall median duration of 165 min from PNBD to furrow ingression (IQR = 63.75; Fig. 4b). We further compared this timing directly to a cohort of 304 EmbryoScope timelapse movies of clinical embryos from patients undergoing fertility treatment, as these fiducial events are visible and comparable (Fig. 4b). Clinical embryos were analysed in four groups: single embryo transfer resulting in pregnancy (defined as a foetal heartbeat on 7-week scan), single embryo transfer resulting in no pregnancy, and non-transferred embryos with either normal (to 2-cell) or abnormal first cytokinesis, from the same set of patients who became pregnant. We would expect our research embryo cohort to have more variation as these have not been retrospectively grouped based on quality. However, the research embryo timing was similar to all the clinical embryo population (Fig. 4b). This further suggests that chromosome imaging does not significantly perturb progression of the first embryonic mitosis (P = 0.1391; research embryos (n = 24) vs. clinical embryos (n = 304)). It remains possible that there are small timing differences between these groups, but larger cohorts of human embryos and higher temporal resolution live imaging would be needed to test for significance.

## The second human embryonic mitosis is less error prone

Following the first mitotic division we continued to film a cohort of embryos to capture the second mitosis (Fig. 5a). We observed 24 cases, including egg-sharer embryos, and determined the timing of key cell division events as described above for the first mitotic division (Supplementary Fig. 4; Supplementary Table 3). We found that the PNBD/NEBD to anaphase duration of mitosis 2 was significantly shorter than that of mitosis 1 (105 vs. 135 min, P = 0.001; Fig. 5b). Prometaphase, metaphase and the overall NEBD to 2-cell duration were all shorter (Fig. 3c). Importantly, we observed no clear lagging chromosome during anaphase of mitosis 2, and only three multipolar divisions (Fig. 5d). This result further demonstrates that errors observed in the first division are highly unlikely to be a consequence of imaging and culturing conditions. While errors appear common during the first mitosis, the second mitosis

seems to have higher fidelity (54% vs. 13.6% total errors, P = 0.0015, Fig. 5d).

## Chromosome segregation errors in the first mitosis are compatible with live birth

Thus far, our data infer that clinical populations may include embryos with mitotic errors, as we saw a high frequency of research embryos displaying chromosome segregation errors in the first mitosis. While we cannot visualise such errors in clinical image sequences, we can investigate their outcomes by examining nuclear morphologies at the 2-cell stage as a proxy (Fig. 6a and Supplementary Movies 10–12). It is well established that lagging chromosomes and multipolar segregation can lead to the formation of micronuclei (as shown in Fig. 2d) and/or multinucleated daughter cells[42,43]. We therefore counted the number of nuclei in the same cohort of 80 clinical time-lapse movies in Fig. 4b, where all embryos were transferred singly and gave rise to clinical pregnancies, defined by foetal heartbeat detection at 7 weeks. We found that 24% of blastomeres in clinical embryos (n = 145, some were unclear) had a variant nuclear configuration at the 2-cell stage (Fig. 6b). Strikingly, when blastomeres from the same clinical embryos were quantified at the 4-cell stage (n = 268), only 1.4% had a variant nuclear configuration (Fig. 6b, second bar). This is consistent with our analysis of the second mitotic division by live-embryo chromosome imaging, where we found a reduction in the number of lagging chromosomes and multipolar divisions (Fig. 5d) and Supplementary Table 3. We then measured the diameter of all nuclei in each blastomere at the 2-cell stage (following the first mitosis) and separated the data into 1 μm bins (Fig. 6c). Plotting the data for 2-cell embryos clearly showed a broad variation in nuclei size with two populations: The first major population had a median diameter of ~24 μm and reflects those found in normal mononucleated blastomeres. The second had a median diameter of 14.9 μm, ranging from 5.8 to ~20 μm. The nuclear variants smaller than 10 μm would be consistent with the micronuclei that form around lagging chromosomes. Such nuclei have been observed previously in Day 3 embryos and shown to contain DNA, with analysis of their blastomeres confirming aneuploidy[24]. We also find that the frequency of these micronuclei in this cohort of human embryos is 12.5% (Fig. 6d). This is in the same range as the number of micronuclei generated from lagging chromosomes that we found in live-embryo chromosome imaging (14.8%, Fig. 2b, d). Thus, the heterogeneity of nuclear phenotypes seen in 2-cell clinical embryos is broadly consistent with the frequency of lagging chromosomes and multipolar divisions seen in egg-share-to-research and deselected embryos. Finally, we investigated pregnancy outcomes in the same cohort of clinical embryos where nuclei were visible in both blastomeres (n = 72), finding that five pregnancies resulted in miscarriage after 7-week foetal heartbeat detection., There was no difference in miscarriage between mononucleated or nuclear variant embryos (8.8 vs. 3.5%

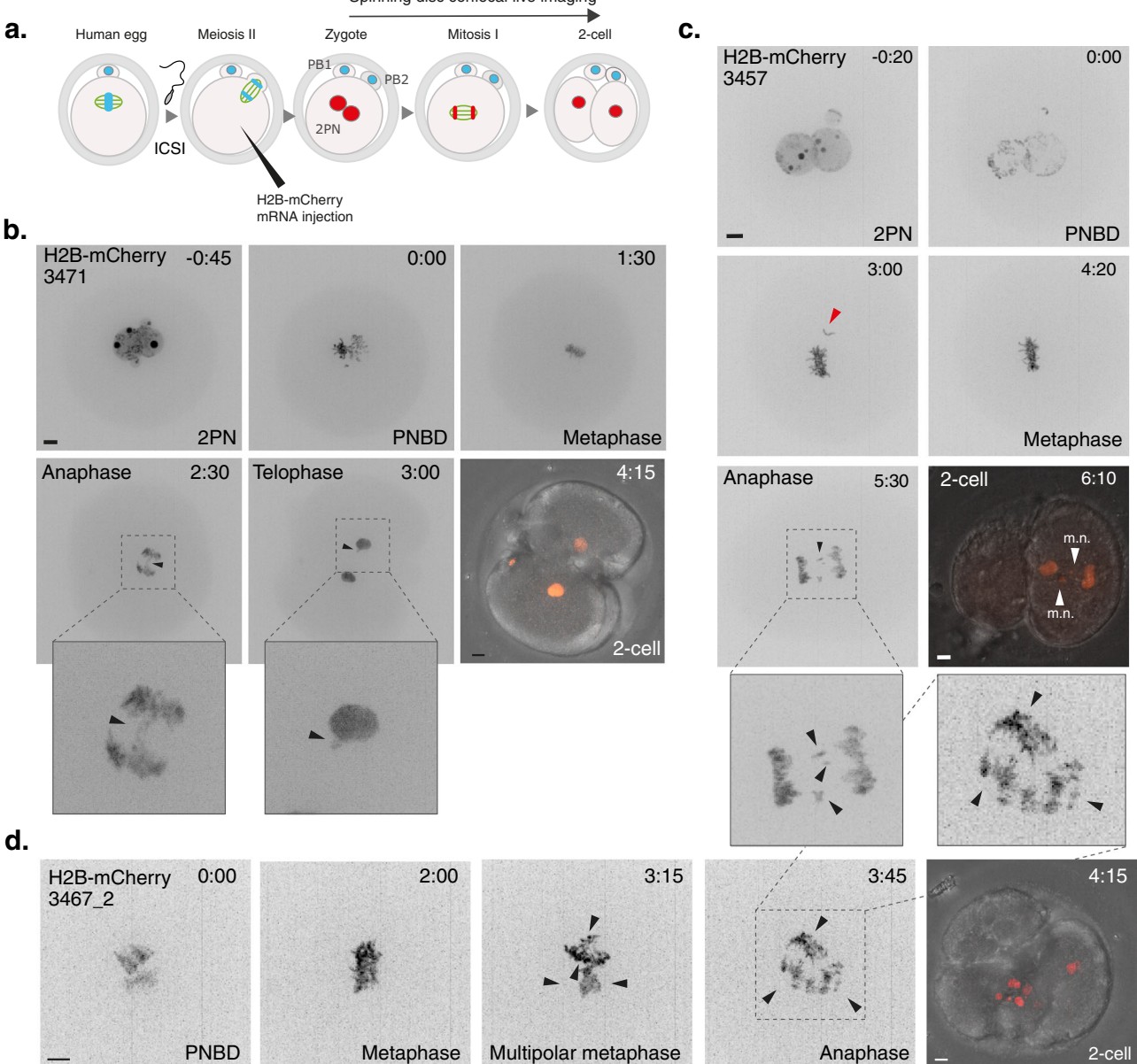

**Fig. 3 | mRNA injected human embryos show errors consistent with SiR-DNA treatment. a** Schematic illustrating key steps for mRNA injection into human 0PN embryos. **b** Time lapse imaging of embryo 3471 progressing through the first mitotic division following mRNA injection. Chromosomes are visualised by H2B-mCherry expression. Scale bar 10 µm, time in hours:mins. H2B-mCherry intensity for merged image is visualised in a non-linear fashion for illustration purposes only. Zoom panels and arrows show lagging chromosomes at anaphase which persist at telophase. **c** Same as **b**, embryo 3457 that progressed through the first mitotic division after aligning the last chromosome (denoted with red arrow), but with multiple anaphase lagging chromosomes that go on form clear micronuclei (m.n.). Merged image shows DNA masses going to one blastomere suggestive of problems in cleavage furrow positioning. **d** Same as **b**, embryo 3467_2 that first aligned chromosomes in a single metaphase plate but then progressed through the first mitotic division by multipolar chromosome segregation with clear lagging chromosomes, forming aberrant nuclei in the merged image. Arrows denote multiple perceived spindle poles.

respectively, *P* = 0.64) (Fig. 6e). Importantly, 96.4% of nuclear variant embryos in the cohort resulted in healthy live birth.

## Discussion

Our data document error-generating phenotypes during the first mitotic division in fertilised human embryos (zygotes). This is indicative of an elevated incidence of chromosome mis-segregation, with our data further suggesting that this can be compatible with live birth. Furthermore, mitotic errors were consistent between embryos discarded from fertility treatment cycles and gold-standard egg-share-to-research embryos. This finding supports the careful use of deselected human zygotes for investigations into human embryogenesis and implantation and suggests that our data are representative of the general embryo population. Finally, we found that the second mitosis had significant reduction in error-associated phenotypes compared to the first. This supports the idea that the first division is uniquely error prone and makes the dominant contribution to preimplantation mosaicism.

Figure 7 outlines the pathways by which lagging chromosomes arising during the first mitosis of a presumed euploid zygote can lead to a mosaic aneuploid embryo. If the lagging chromatid forms a micronucleus and/or mis-segregates to the wrong daughter

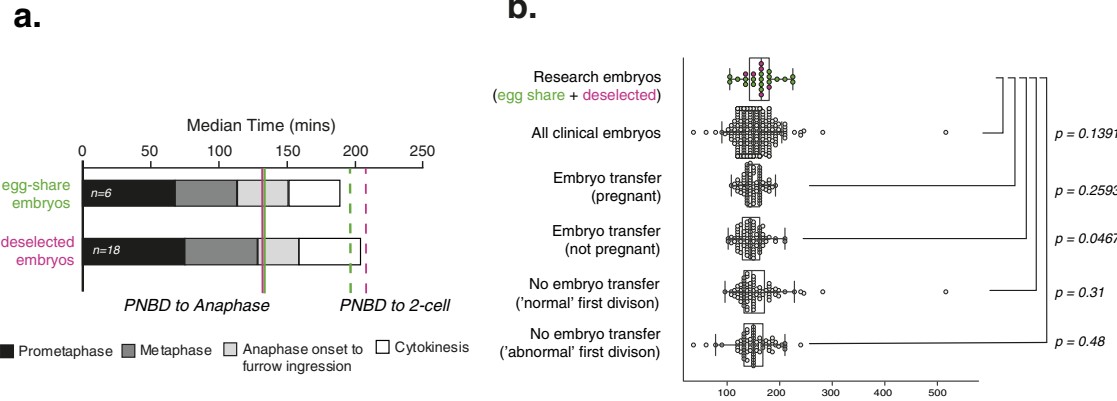

**Fig. 4 | Deselected embryos have similar mitotic timing to clinical embryos.**
**a** Median durations of each mitotic phase plotted consecutively from 6 complete egg-share embryos movies and 18 deselected embryo movies during the first mitosis. **b** Quantification of timing from PNBD or pronuclear fading to furrow ingression (start of cytokinesis) for complete movies of research embryos (deselected (pink) and egg-share (green), $n = 24$) and clinical embryos (imaged using EmbryoScope during patient treatment, grey background, $n = 304$). Box and whisker plots represent minimum, lower quartile, median, upper quartile and maximum. Outliers fall beyond these lines. Individual data points are overlaid. Clinical embryos fall into four categories: singly transferred embryos which gave rise to clinical pregnancy, singly transferred embryos which did not give rise to clinical pregnancy, non-transferred embryos which divided normally into 2 cells, and non-transferred embryos which divided into 3 or more cells in the first division. 95% confidence intervals (bottom-top, min): Research 149.61–176.63, All embryos 146.57–154.38, transferred pregnant 144.14–152.4, Transferred not pregnant 139.86–150.19, Non-transferred normal first division 146.06–169.24, Non-transferred abnormal 142.55–158.86. Whole data $P$ value from the Kruskal–Wallis test is 0.1151. Pairwise $P$ values from Kruskal–Wallis test with post hoc Dunn's test. Source data are provided as a Source data file.

(nondisjunction) then monosomic and trisomic blastomere lineages will be created. However, if the lagging chromosome were lost, for example, via chromothripsis, then the outcome would be a mosaic embryo with a mix of monosomic and euploid lineages. Euploid cells could also be created through mitotic rescue of the trisomic or monosomic blastomere through chromosome gain or loss in subsequent divisions—which can create uniparental disomy, although this is rare[13]. These mosaic patterns are consistent with those detected using single cell RNA sequencing and array-based methods in dissociated blastocyst stage embryos[15,16].

More complex aneuploidies can be explained by multipolar chromosome segregation, which was also observed during the first mitotic division. Indeed, hypodiploid chromosome complements—a signature of tripolar mitosis—have been identified in day 3 human embryos[36]. Furthermore, Vanneste et al.[16] showed that human blastomeres often carry a high frequency of structural aneuploidies in the form of segmental deletions, duplications, and amplifications. We can speculate that these may arise from lagging chromosomes whose fate is a micronucleus: in somatic human cell lines and cancers such chromosomes are subject to defective DNA replication and chromothripsis which can generate significant chromosomal rearrangements[40].

Our study does not exclude the possibility of further segregation errors arising after the first two divisions, and previous work has identified examples of multipolar spindles and lagging chromosomes at various developmental stages[23,25]. Such later mis-segregation events would be required in order to explain the wide variation in the number of aneuploid blastomeres between embryos[15]. For example, an error occurring at the 8-cell stage would create a smaller aneuploid lineage than at the 2-cell stage. We detected no clear lagging chromosomes in the second mitotic division (Fig. 5d), and nearly all 4-cell stage clinical embryos showed normal nuclear morphology (Fig. 6b). These data suggest that the error rate is reducing as embryogenesis proceeds; however, future work with larger numbers of embryos is needed to establish precise error rates in later mitotic divisions.

Why is the first mitotic division so error prone? Lagging chromosomes are most likely the result of merotelic attachments where a kinetochore retains microtubule attachments to both spindle poles as anaphase initiates[35]. These erroneous attachments may be the result of inefficient error correction of mal-oriented kinetochores and/or defects in spindle geometry that increase probability of improper kinetochore attachment[44]. Recent results in mouse and bovine zygotes point to the latter, as dual mitotic spindles are observed to form separately around each parental genome[45,46]. We postulate that failure to organise a dual spindle would increase the risk of multipolar chromosome segregation and frequency of erroneous kinetochore-microtubule attachments. Consistent with this, we were able to observe examples of where chromosomes appear to be segregated by two distinct spindles in mis-fertilised human zygotes (Supplementary Fig. 2b). Furthermore, efficient pronuclei migration and chromosome clustering at the interface also limits lagging chromosomes in bovine zygotes[28]—which occur at a similar frequency to that reported in our human embryos. How pronucleolar clustering relates to clinical outcomes in human embryos is unclear[47], however suboptimal clustering could be a mechanism contributing to chromosome segregation errors and mosaic aneuploidy arising in the first human mitotic division. Higher-resolution imaging data will be required to address this. Importantly, the sheer scale of the human zygote (~115 μm diameter) in comparison to a typical mitotic spindle (~15 μm) may also be a factor; this could negatively affect spindle assembly checkpoint (SAC) fidelity due to dilution of mitotic checkpoint complex in the large cytoplasm, if the SAC is indeed active in human embryos. This has been demonstrated in *C. elegans* embryos[48], but appears not to be conserved in rodents[49]. Further work is needed to evaluate the contribution of SAC signalling to human embryonic aneuploidy.

Understanding the origins and consequences of mosaic aneuploidies may provide insight into the causes of human pregnancy loss and improve the success of assisted reproduction technology (ART)[50]. Indeed, the exclusion of mosaic aneuploid embryos from patient treatment is not advisable[17,18]. Our data provide further evidence that errors in the first mitosis are compatible with live birth. The human fertility success rate averages ~30% per cycle in both natural conception and ART, suggesting that correction or rescue of mosaics (e.g. refs. 13, 51, 52) is of critical importance during human development and warrants further study.

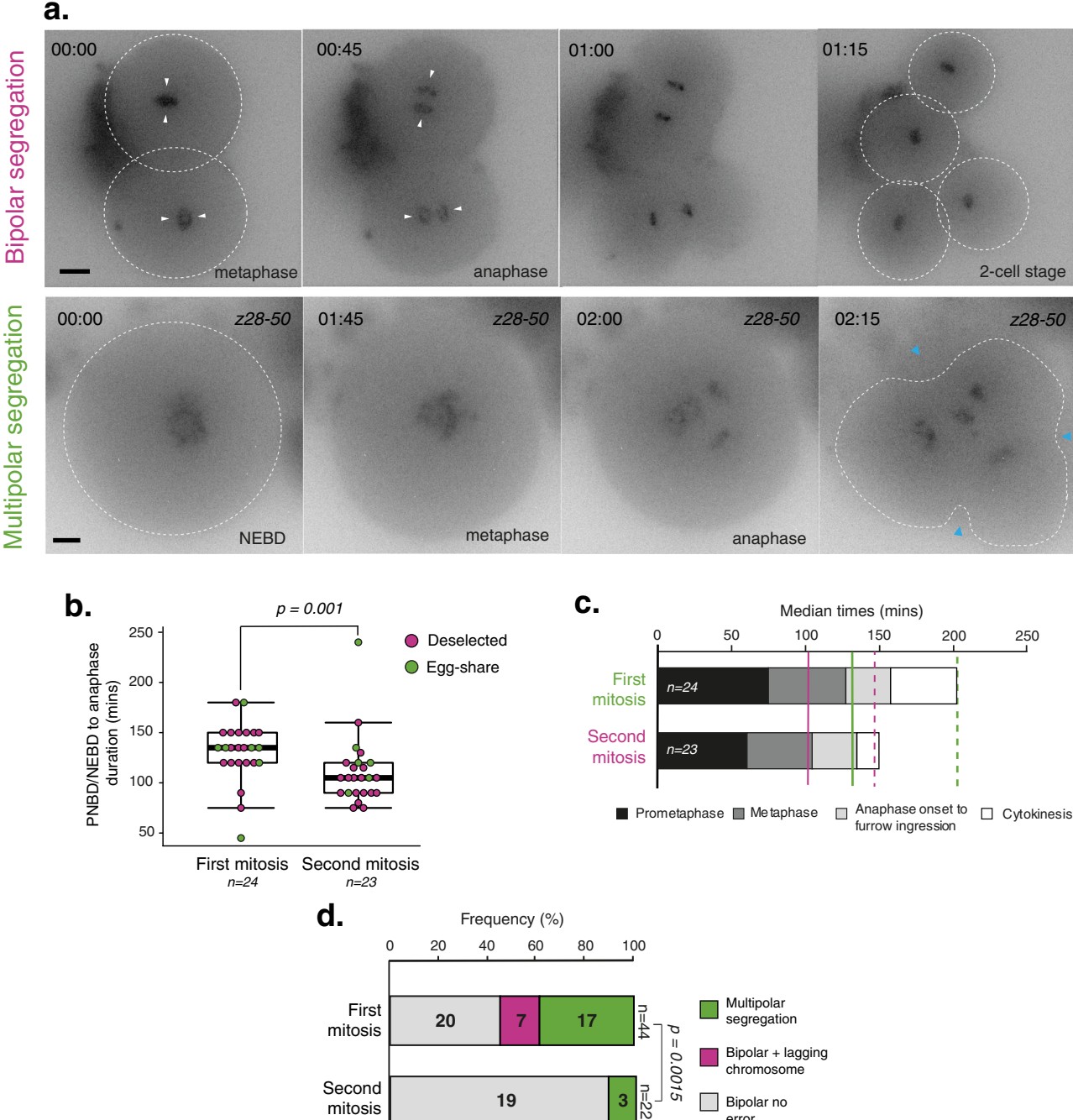

**Fig. 5 | Comparing the first and second mitotic divisions of human embryos.**
**a** Time lapse imaging of representative deselected human embryos progressing through the second embryonic mitosis with bipolar division (top) and multipolar division (below). White arrows indicate bipolar segregation and white circles indicate cell outline. Chromosomes are visualised using SiR-DNA dye, time in hours:mins. Blue arrows indicate onset of cleavage furrow ingression, scale bar 20 μm. **b** Quantification of the timing from PNBD/NEBD to anaphase onset of deselected and egg-sharer embryos undergoing the first or second mitosis. *P* value from a two-sided Mann–Whitney *U*-test. *N* refers to the number of filmed embryos included in dataset. Box and whisker plots represent minimum, lower quartile, median, upper quartile and

maximum. Outliers fall beyond these lines. Individual data points are overlaid. **c** Median durations of each mitotic phase plotted consecutively to compare complete mitosis 1 and mitosis 2 movies, including both egg-share and deselected embryos. Solid lines indicate median NEBD to anaphase onset duration, dashed lines indicate median NEBD to 2/4-cell duration. **d** Quantification of anaphase errors (multipolar chromosome segregation and lagging chromosomes), occurring during the first mitosis (*n* = 44) or second mitosis (*n* = 23) in deselected and egg-share embryos. Chromosome segregation errors could not be accurately quantified in one mitosis 1 and 2 division respectively, so these were excluded from this analysis. *P* value from a two-sided Fishers exact test. Source data are provided as a Source data file.

## Methods

### Donation of human embryos to research

The NHS Research Ethics Committee approved both the research project (Indicators of Oocyte and Embryo Development, 04/Q2802/26) and egg-sharing-to-research programme (19/WM/0003). All work

was conducted under a Research Licence from the Human Fertilisation and Embryology Authority (HFEA; R0155; Indicators of Oocyte and Embryo Development). Informed consent for donation of eggs, embryos and sperm to research was provided voluntarily, at an appointment with a research nurse, in advance of their treatment by

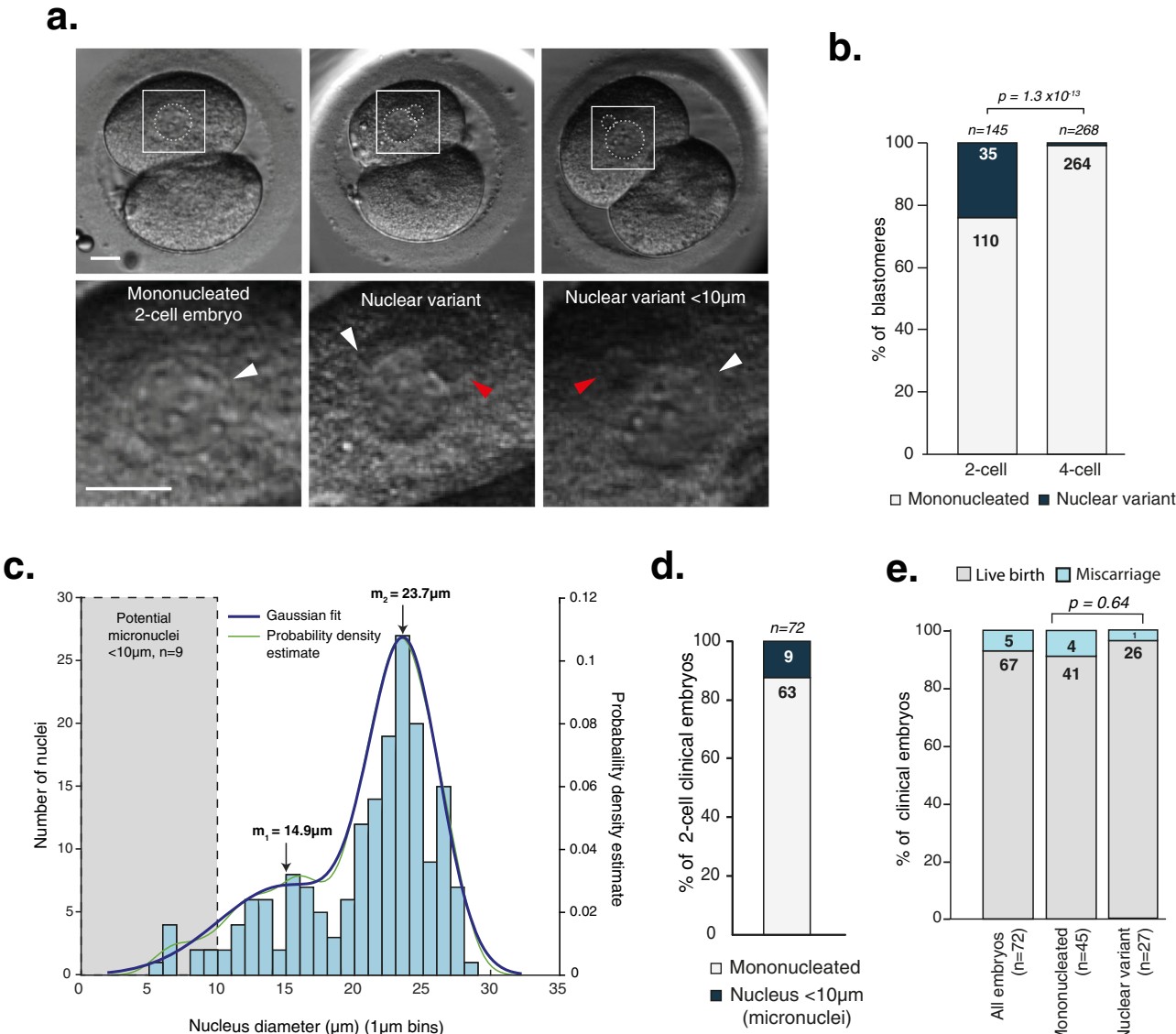

**Fig. 6 | Nuclear defects arising in the first embryonic mitosis are compatible with live birth. a** Clinical EmbryoScope movie stills of 2-cell clinical embryos which gave rise to pregnancies (foetal heartbeat detected), with different nucleation status. Nuclei are outlined in white. Scale bars 20 μm. White arrows label 'normal' sized nuclei, red arrow labels a nuclear variant. Further example movies are provided on OMERO, see methods for details. **b** Quantification of blastomeres in 2-cell clinical embryos which are either mononucleated or displayed nuclear defects (first bar). The nucleation status of these same embryos was quantified at the 4-cell stage (second bar). *P* value from a two-sided Fishers exact test. **c** Histogram showing size distribution of all nuclei measured in 2-cell embryo blastomeres, grouped into 1 μm bins. Gaussian distribution plotted using MATLAB Ezyfit Toolkit: Output parameters: a_1 = 0.028372, a_2 = 0.10168, m_1 = 14.964 μm, m_2 = 23.71 μm, s_1 = 4.8823 μm, s_2 = 2.5808 μm **d** Quantification of the number of 2-cell embryos containing nuclei <10 μm (micronuclei). Only embryos where nuclei in both blastomeres could be quantified are included. **e** Quantification of pregnancy outcomes from the same group of clinical embryos related to phenotype at the 2-cell stage (only embryos where both blastomeres could be measured). Miscarriage occurred after foetal heartbeat detection at 7 weeks. *P* value from a two-sided Fishers exact test. Source data are provided as a Source data file.

patients undergoing in vitro fertilisation (IVF) or intracytoplasmic sperm injection (ICSI) at the Centre for Reproductive Medicine (CRM), University Hospitals Coventry and Warwickshire (UHCW) NHS Trust or at the Edinburgh Fertility and Reproductive Endocrine Centre (EFREC). Details of the clinic success rates are published by the Human Fertilisation and Embryology Authority (https://www.hfea.gov.uk/choose-a-clinic/clinic-search/results/13/). Participation in research was optional and did not affect the treatment that the patient received. Patients were aware of the purpose of the research. Standard clinical protocols were used including ovarian stimulation with FSH preparations according to either GnRH agonist or antagonist regimens, and the Origio suite of culture media. None of the embryos was subject to preimplantation genetic testing or sex determination. The deselected

material used for research was unsuitable for the patients' treatment, due to delay (0PN) or mis-fertilisation (1PN/3PN), and would otherwise have been disposed of. 1PN embryos may be haploid, but if diploid, may arise from fusion of the parental pronuclei and can lead to live birth[53]. 3PN embryos usually form by fertilisation with two spermatozoa, or polar body extrusion failure[54]. They are associated with spontaneous abortions, however a euploid live baby has been born from a tripronuclear embryo[55]. Patients providing material unsuitable for use in their treatment received no compensation.

Egg-share material was collected from volunteer female patients aged ≤33 on the egg sharing to research programme at the CRM only. These patients voluntarily elect to share half of their eggs with the research programme, and in return receive their treatment for a

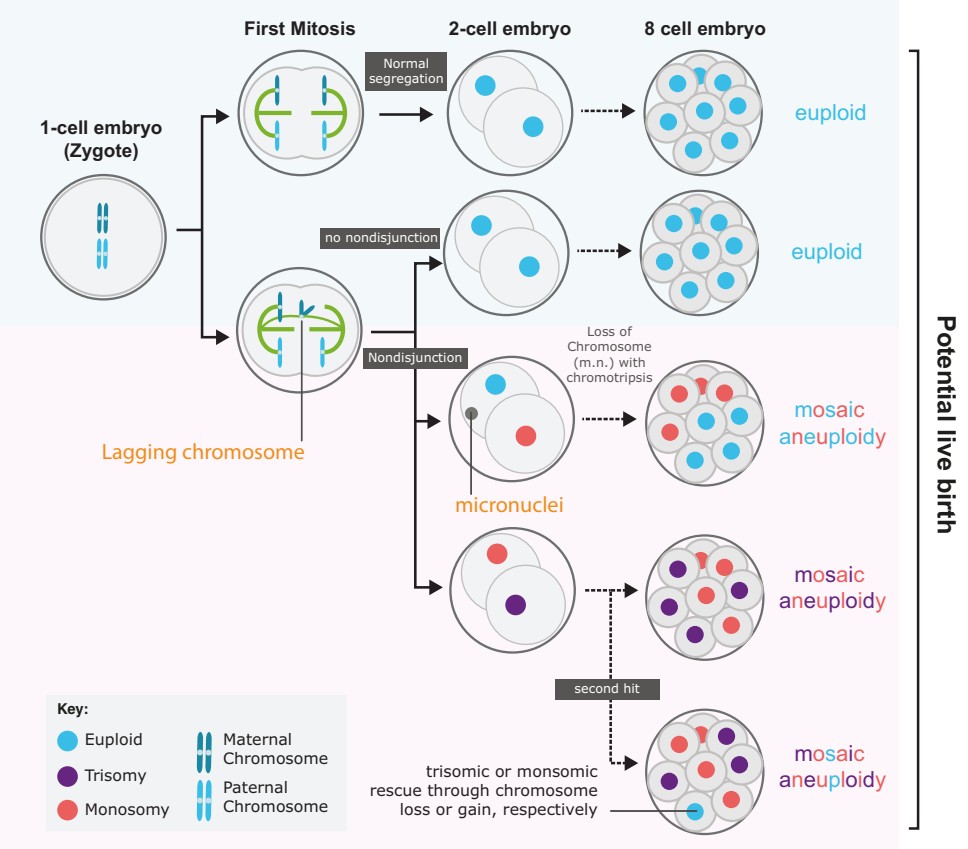

**Fig. 7 | Model of how first division errors can lead to different euploid/aneuploid embryos.** Top row, a pair of homologous chromosomes (blue) are correctly segregated into the 2-cell embryo. Subsequent divisions (dotted black arrow), if occurring without error, would lead to an 8-cell embryo in which all blastomeres are euploid (blue). Second row; Non-disjunction during the first mitosis would result in trisomic and monosomic blastomeres (purple and red) with two potential outcomes: upper−lagging chromosome becomes a micronucleus (m.m.) that undergoes chromothripsis and potential chromosome loss. Outcome is euploid (blue) and monosomic (red) lineages in 8-cell (and beyond) embryo. Lower− without chromosome loss the outcome is a mosaic embryo with monosomic and trisomic lineages (purple and red). Such aneuploid lineages may be rescued in later divisions through further 'second hit' non-disjunction, possibly resulting in euploid blastomeres with a uniparental disomy signature.

reduced cost (£300 for IVF or £770 for ICSI, rather than the standard cost of £2800 (IVF) or £3250 (ICSI), excluding medication), subsidised by the research funder. This is in keeping with UK law and approved by the NHS Research Ethics Committee and the Human Fertilisation and Embryology Authority, referenced above. Patients recruited into the egg share programme progressed through their IVF cycle in the normal way, but immediately after egg collection, their cumulus-enclosed eggs were allocated randomly to either their own treatment or to research. In the event of an odd number of eggs being collected, the patient received one more egg than the research project. The patient's own treatment proceeded routinely using half the original number of eggs. The eggs allocated to research were denuded of cumulus cells. Mature (metaphase II) eggs were inseminated with fertile donor sperm via ICSI using standard clinical protocols, in order to create embryos for research use in this project. One sperm donor was used for all research inseminations. Following sperm injection, eggs were incubated in an Embryoscope™ time lapse incubator, in an identical manner to embryos used clinically. Immature eggs collected at the same time from egg sharers were used for research not included in this publication.

## Chromosome imaging of human embryos

Embryos arising from the egg-share programme were collected from the clinic 1−2 h after the fertilisation check had been completed by a clinical embryologist, -17 h after insemination. Deselected embryos were collected from the clinic around 3−5 h after the fertilisation check. Thus, none of the embryos that we used were cryopreserved. For SiR-DNA

treatment, the zona pellucida of individual embryos was removed by brief incubation and pipetting in prewarmed acid Tyrode's solution (Sigma). Embryos were transferred to a Fluorodish (WPI) containing 2 μM SiR-DNA (Spirochrome) diluted in prewarmed Cleav medium (Origio) under mineral oil. Embryos were transported -14 km from UHCW CRM to Warwick Medical School (WMS) in a portable incubator (K Systems) held at 37 °C, and transferred to the microscope immediately upon arrival. Image stacks (60 × 1.5 μm optical sections; 1 × 1 binning) were acquired every 10 or 15 min for a 24−36 h period. The embryos were imaged with a ×40 oil-immersion 1.3 NA objective (Olympus) using a DeltaVision Elite microscope (Applied Precision, LLC) equipped with a CoolSNAP HQ camera (Roper Scientific). Fluorescent images were acquired using an InsightSSI solid state illuminator (Applied Precision, LLC) attenuated to 32% and a Cy5 filter set with an exposure time of 0.05 s. Brightfield images were acquired attenuated to 10% with an exposure time of 0.1 s. A stage-top incubator (INU; Tokai Hit) maintained embryos at 37 °C and 5% $CO_2$ with further stabilisation from a microscope enclosure (Weather station; Precision Control) held at 37 °C. The temperature was confirmed with a calibrated probe (Fluke 52). Image sequences were inspected and analysed by hand using OMERO (Open Microscopy Environment). Embryos from egg-sharer patient 3371 and mRNA injected embryos were imaged using Marianas spinning disk confocal microscope from 3i (Intelligent Imaging Innovations) equipped with 2× Photometrics 95B Prime sCMOS cameras, using a ×40 oil-immersion 1.46, alphaPlnApo (Zeiss) objective. Fluorescent images were acquired every 15 min (60 × 1.5 μm optical sections) with a 640 nm laser

attenuated to 5% with an exposure time of 50 ms and brightfield images were acquired attenuated to 10% with an exposure time of 40 ms. A stage-top incubator (Okolab) maintained embryos at 37 °C and 5% $CO_2$. Temperature and $CO_2$ were verified using calibrated probes.

### mRNA synthesis and injection

For mRNA injections, the T7-H2B-mCherry plasmid (Addgene 20972) was linearised with SmaI. Capped mRNA was synthesised using mMessage mMachine™ T7 transcription kit (Invitrogen) and diluted to a concentration of 1 μg/μl. Zygotes were microinjected using Eppendorf FemtoJet® 4i and Femtotip II microinjection capillaries with the following parameters; Injection pressure 150 hPa, compensation pressure 35 hPa, injection time 0.6 s, and incubated at 37 °C and 5% $CO_2$ for 3 h before start of imaging. Fluorescent time lapse movies were acquired using a 3i Marianas spinning disk confocal microscope as described above, except using the 561 nm laser attenuated to 3% with an exposure time of 60 ms and 48 × 1.5 μm optical sections.

### Immunofluorescence

Zygotes were monitored for pronuclei break down, then washed through warmed PHEM buffer (60 mM PIPES, 25 mM HEPES, 10 mM EGTA, 4 mM $MgSO_4.7H_2O$; pH 6.9) with 0.25% Triton X-100 at 37 °C, fixed with 3.7% paraformaldehyde in PHEM buffer with 0.25% Triton X-100 for 30 min and permeabilised in PBS with 0.25% Triton X-100 for 15 min at room temperature. Embryos were stored in PBS with 0.05% Tween-20 (PBST) until immunofluorescence was performed. For immunofluorescence, embryos were blocked in 3% BSA in PBST at 4 °C overnight, then incubated with antibodies against α-tubulin (Mouse 1:200; T6074, Sigma) and CENPC (guinea pig 1:200; MBL, PB030) in 3% BSA PBST at 4 °C overnight. Embryos were then washed for 3 × 20 min in 1% BSA PBST and further incubated with secondary antibodies; goat anti-mouse Alexa Fluor 555 (1:500; A-21422, ThermoFisher Scientific) and goat anti-guinea pig Alexa Fluor 647 (1:500; A-21450, Thermo-Fisher Scientific). Embryos were mounted in ProLong™ Gold Antifade Mountant with DAPI (P36931, ThermoFisher Scientific) on a FluoroDish (FD35-100, WPI). Samples were imaged using LSM980 laser scanning confocal equipped with an Airyscan detector (Zeiss UK, Cambridge) using a Plan-APO (×63/1.4 NA) oil objective (Zeiss). A 0.14 μm optical section spacing was used to encompass the area of interest. Also, 405, 561 and 639 nm lasers were used to detect DAPI staining and Alexa Fluor 555 and Alexa Fluor 647, respectively. Images were prepared using Fiji (National Institutes of Health).

### Clinical imaging and analysis of human embryos

A cohort of 304 clinical human embryos were analysed in this study. These embryos were imaged on an EmbryoScope™ as part of patient ART treatment. Hoffman modulation contrast images are collected every 10 min for up to 6 days (blastocyst formation). These embryos fall into 3 categories: 80 singly transferred embryos which gave rise to clinical pregnancy, 76 singly transferred embryos which did not give rise to clinical pregnancy and 148 embryos (from the 80 patients who had single embryo transfers and became pregnant) which were not transferred or cryopreserved due to abnormal morphology (80 which had a normal first cytokinesis, dividing into 2 cells and 68 which had an abnormal cytokinesis, dividing into 3 or more cells). Clinical pregnancy is defined as foetal heartbeat detection at 7 weeks. The timing of events (PNBD, the start of cytokinetic furrow ingression and the appearance of 2 distinct cells) during the first cell division were determined and presented in Fig. 4b. Nuclear phenotype was further assessed in blastomeres of 80 embryos as they progressed from the 2-cell to the 4-cell stage and presented in Fig. 6. These are the same 80 embryos that gave rise to clinical pregnancy in Fig. 4b. If nuclei were not visible in all blastomeres, the embryos were excluded from analysis in Fig. 6d. Each blastomere was visually examined for nuclear variants using multiple-focal planes from time lapse sequences. A FIJI macro was used to

compile embryoscope data files (https://github.com/Laura190/folders2s). The diameter of all visible nuclei was measured using FIJI line tool and converted from pixels to μm in Fig. 6c.

### Data and statistical analysis

Mann–Whitney U-tests for Fig. 5b, Fisher's exact tests for Figs. 2b, c, 5d, e, and 6b, e, and Kruskal–Wallis test with post hoc Dunn's test for Fig. 4b were performed using MATLAB R2020A (Mathworks) inbuilt functions. Gaussian distribution in Fig. 6c was plotted using EzyFit 2.44 toolbox. Equation: $y(x) = a\_1*exp(-(x-m\_1)^2/(2*s\_1^2))+a\_2*exp(-(x-m\_2)^2/(2*s\_2^2))$[56].

### Reporting summary

Further information on research design is available in the Nature Research Reporting Summary linked to this article.

## Data availability

The metadata archive is hosted on Zenodo: https://doi.org/10.5281/zenodo.7075621. This links to the source data used in this study, which are available in the Warwick OMERO database. These can be found via this link using OMERO.web viewer: https://warwick.ac.uk/fac/sci/med/research/biomedical/facilities/camdu/publicdata/, selecting the link to the dataset for this paper (Currie et al., 2022). Login and password: both 'public'. The analysed data generated in this study are provided in the Source data file. Requests for clinical imaging data should be made to G.M.H. to ensure compliance with NHS research governance for patient confidentiality and ethical approvals. Source data are provided with this paper.

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

## Acknowledgements

We are hugely thankful and indebted to the patients, embryologists, research nurses and medical consultants at the Centre for Reproductive Medicine, University Hospitals Coventry and Warwickshire NHS Trust. Special thanks to all Warwick and Edinburgh colleagues in the Eggs 'n Embryos research group for insightful and fun discussions—supported by a Wellcome Collaborative Award (215625). We also gratefully acknowledge CAMDU (Computing and Advanced Microscopy Unit) for their support and assistance in this work, particularly Claire Mitchell for microscopy assistance and Laura Cooper for data visualisation and data management. We also thank the Centre Optical Imaging Laboratory and David Kelly at the Wellcome Centre for Cell Biology for microscopy support. We thank Aleksandra Byrska for data validation and Lina Germanova for help with MATLAB. A.D.M., A.L.M., G.M.H., C.E.C., B.M. and D.T. are supported by a Wellcome Collaborative Award (215625). E.F. is supported by a Warwick Collaborative Postgraduate Research Scholarship with UHCW. A.D.M. is also supported by a Wellcome Senior Investigator Award (106151) and a Wolfson Royal Society Research Merit Award (WM150020). G.M.H. is also supported by the WPH Charitable Trust. A.L.M. is also supported by a Wellcome Investigator award (220780) and core funding for the Wellcome Centre for Cell Biology (203149).

## Author contributions

C.E.C.: data curation, reviewing and editing, validation, investigation (established methods to inject human embryos with Histone2B-mCherry mRNA and carried out the live imaging), live imaging of egg sharer embryos with SiR-DNA, analysis of SiR-DNA, mRNA and embryoscope data in Figs. 4 and 6, E.F.: investigation (established and carried out live imaging of deselected and egg sharer embryos using SiR-DNA), analysis of SiR-DNA and embryoscope data in Figs. 4 and 6, editing. L.B.W.: analysis of embryoscope data in Fig. 6, D.M.T.: methodology, resources, B.M.: investigation (super-resolution imaging of mitotic spindles in human zygotes), M.E.: data curation and analysis of SiR-DNA imaging data A.L.M.: conceptualisation, supervision, reviewing and editing, funding acquisition, G.M.H.: conceptualisation, writing, reviewing and editing, supervision, ethical approvals, funding acquisition, A.D.M.: conceptualisation, supervision, formal analysis, original draft preparation, writing and editing, funding acquisition.

## Competing interests

The authors declare no competing interests.
