## [Peer Review File · Nature Communications]

The first mitotic division of human embryos is highly error proneREVIEWER COMMENTS

Reviewer #1 (Remarks to the Author):

In the manuscript 'The first mitotic division of human embryos is highly error prone', the authors use advanced imaging techniques using a nuclear stain to visualize chromosome segregation errors in the first and second cell divisions in human embryos. These observations are made on embryos generated from donated oocytes and embryos deemed unsuitable for patients' treatment due to delayed or mis-fertilization. The authors also analyze images from clinical time-lapse cases, visually evaluating signs of mitotic errors.

Aside from its biological significance, the topic is clinically relevant, considering that early mitotic errors are the source of chromosomal mosaicism in human embryos; there is an ongoing intense debate on how embryos with evidence of mosaicism should be managed in the IVF clinic.

The strong points of the paper are that 1) a method has been set up to visualize live chromosome dynamics in early human embryos, and 2) some of the proposed mechanisms giving rise to mosaicism (previously observed in other organisms) do in fact play a role in human embryos (anaphase lag and tripolar mitosis) and 3) these errors are much more common during the first cell division than during the second cell division.

The manuscript is well written, the references are appropriate, and the figures are well designed and clear .

As soon as the authors start making claims about the incidence of mitotic errors during early divisions in human embryos, they need to tread very carefully. Their sample size of egg-share embryos is admittedly small (understandably, those are very hard to come by), and the population of deselected embryos is not representative of the general embryo population, being highly selected for abnormalities. This is why I consider it a stretch to claim that the first mitotic division is generally 'highly error prone'. In truth, very little can be said from these experiments about the overall proportion of embryos with early mitotic errors. I think the authors might consider softening their language in the text when discussing the incidence/frequency/percentage of early mitotic errors in the general embryo population (for example, the first sentence of the Discussion, but also recurrently in the text). On the other hand, the experiments certainly suggest that the first mitotic division is significantly more error prone relative to the second mitotic division, which is a point that should be emphasized.

Major Points:

- I was not able to access any of the original Movies. There are plenty of representative movie stills in the Figures, but I would like to see actual clips of the time lapse imaging with SiR-DNA stain, as well as the clinical movies/Embryoscope, showing some of the mitotic events discussed in the manuscript. Such visuals would definitely elevate the paper.
- Using SiR-DNA the authors can visualize processes which they interpret as mitotic chromosome segregation errors, but they never actually confirm such events by molecular means. For example, if an anaphase lag is apparently observed at the first division, could the authors do chromosomal testing of the two daughter cells (or even later, at the four-cell stage) to confirm that indeed there is a chromosome number error in one or more cells? This can be done by common PGT-A platforms, as I don't see how SiR-DNA treatment could interfere with PGT-A chemistries. I think such a validation would considerably increase the confidence in their system consequently in the conclusions the manuscript makes.
- Regarding the clinical samples visualized by Embryoscope, the authors hypothesize that the small nuclei observed in some samples are micronuclei that form around lagging chromosomes. Again, this could be substantiated by molecular methods and showing that indeed daughter cells contain

chromosome copy number errors. By the way, confirming a correlation between small nuclei and aneuploidy would be quite impactful. Many IVF clinics use time-lapse imaging such as Embryoscope, and this could be a way to quickly and easily predict aneuploidy in embryos.

- Figure 4a: assessment of nuclei seems highly subjective, at least judging from the images shown. If the authors have other sample images of better quality, they should consider exchanging them. Was there any software or standardized method used to determine if a nucleus was 'large', 'medium', or 'small'? It says in the methods that a trained embryologist evaluated the nuclei. But my concern is that some of those dots might not even be micronuclei. It would be more convincing if the authors could take some of those embryos with visual evidence of micronuclei by Embryoscope and do a nuclear stain, showing that the small nuclei and/or 'micronuclei' do indeed contain chromosomal material.

- The findings might not be entirely representative of clinical-use embryos (let alone embryos from natural conceptions!), since the experimental embryos were exposed to several manipulations (zona pellucida removal by chemical treatment, addition of SiR-DNA, transport to imaging site, and special imaging method). I am also still not convinced that SiR-DNA treatment does not influence the rate/type of mitotic error. The authors mention that overall, the developmental timeframes are similar between treated and untreated embryos, but I don't think stage progression is a sufficiently informative metric regarding mitotic errors and whether their protocol can induce them. For example, I cannot agree with the statement: 'This confirms that chromosome imaging does not perturb the first embryonic mitosis...'. SiR-DNA interaction with DNA could possibly interfere with the kinetochore-microtubule complex, thereby inducing mitotic mistakes. These concerns and caveats should be dispelled with further experiments or at the very least carefully explored in the discussion.

- What is happening between the first and second division that increases the fidelity of chromosome segregation? Why is the first division much more error prone than the second? They mention the ratio of cell size to spindle in the second-to-last paragraph of the discussion, but the authors should elaborate and/or offer additional explanations (since in my mind is the major biological finding of the study).

Minor points:

- Figure 1c: Why was PNBD not observed in 6 out of 13 embryos for research? And yet, it was observed in all 16 embryos used for treatment? In view of those differences, can the following really be claimed: 'suggesting that the fluorescent imaging methodology did not impair embryo progression.' On the contrary, there seems to be a clear difference between the embryos treated with SiR-DNA and the others. Please clarify.

- Figure 1: embryo number 3 of patient 3247 of Figure 1c has an asterisk on it and refers to the embryo shown in 1b. However, I thought that egg-share-to-research embryos were free from the zona pellucida, and yet it is still present in 1b. See also the text on this point: 'In parallel, patient treatment embryos were monitored in the clinic via time-lapse imaging with Hoffman contrast (EmbryoScope™) to inform embryo selection (Fig. 1b).

- Figure 1c: Indicate which embryo(s) are depicted in 1e.

- Figure 1c: What happened to the embryo 4 of patient 3247 in Figure 1c? What about embryo 2 of patient 3371 in Figure 1c? Are those instances where the zygote immediately divides? Please clarify.

- Fig 2: How were the embryos 'unsuitable for transfer' used? Were they frozen at the zygote stage, donated, then thawed and used for the experiment? Or did the patient consent to donate such

embryos ahead of time? I suspect it's the second one, but then, were the zonas removed for this group of embryos as well? Please clarify these points in the manuscript.

- Figure 2 c: Why are 1PN and 2PN grouped together?

- Figure 3 d and e: What does e) tell us that cannot already be deduced from d)?

- Figure 4c: 4 embryos have gone missing 50+18 in the left column but only 64 in the right column. What happened to those 4 cases?

- Figure 4e: My understanding is that nondisjunction and anaphase lag are two different mechanisms, one leading to reciprocal aneuploidies in daughter cells, the other leading to a loss of chromosomes in one daughter cells (see doi: 10.1093/humupd/dmu016). I don't see how the authors are combining both mechanisms in their model (Fig 4e). Their siR-DNA imaging system does not allow the authors to visualize nondisjunction (the extra chromosome going to one daughter cells cannot be observed at this resolution). I don't think any claims can be made regarding nondisjunction based on the data presented here.

- Supplementary Fig 2 a): It's very hard to distinguish anything in the siR-DNA panels (including the high magnifications). Are there any better ones that can be used?

- There are two papers that the authors should consider incorporating in their manuscript as they are highly relevant to the discussion of mosaic embryo management in the IVF clinic:

doi:10.1093/humrep/dez309 and

doi: 10.1016/j.fertnstert.2020.11.041

Reviewer #2 (Remarks to the Author):

In their submitted manuscript, Ford et al. apply a novel approach for visualizing chromosome dynamics during the first two cell divisions of the human embryo, quantifying various aspects of mitotic progression. Overall, this work is interesting and important for the fundamental understanding of human development, which in turn provides a foundation for fertility therapies, etc. The comments that follow largely focus on providing additional context and clarifying the novel aspects of this work, along with some statistical recommendations that do not require additional experimentation.

Major comments:

- This work is clearly novel in presenting a live-embryo chromosome imaging methodology for humans and related to the phenomenon of mosaicism that is of longstanding interest and debate in the field of human IVF. However, I think the novel aspects of this work in relation to past work can be better emphasized and contextualized, perhaps by expanding the Introduction and Discussion. For example, the sentence "Live-embryo chromosome imaging allows us to determine the timing of key chromosome related events that are invisible in clinical image sequences" in the Results implies aspects of mitotic timing that were previously unknown. This could be stated earlier, in the Introduction, and expanded upon to clarify what exactly was known and unknown (in humans; in other organisms, etc.).

- Related to the previous point, can you please clarify for a non-specialist why observation of the chromosomes is necessary for estimating the timing of mitotic events (in contrast to conventional time

lapse imaging)?

- Many of the Results in the study hinge on the observation that abnormally fertilized embryos available to research had similar developmental properties and could be used to draw conclusions about normally fertilized embryos. The evidence provided was that statistical tests comparing the two groups were not significant, with p-values provided. However, it was unclear to me whether in this small sample, you have enough statistical power that a lack of significant differences is meaningful. One way to address this would be to include confidence intervals along with p-values, so that we know how large of a difference we can rule out. Another way to address this would be to more honestly discuss the limitation in sample size and the fact that abnormally fertilized embryos may not be a perfect proxy for fertilized embryos and how this may impact interpretation of your results.

- Related to the point above similar logic was used to argue that the imaging and culture conditions did not impact mitotic fidelity, but this should also be moderated as suggested above.

- Please provide an explanation for why only the first two cell divisions were tracked. Is this a technical limitation (e.g., degradation of the dye used for imaging chromosomes)? Or a practical limitation (tracking more cells makes data collection too challenging)? What do you speculate you would observe if you tracked additional divisions, especially after embryonic genome activation?

Major point-by-point comments:

- Introduction

o In the discussion of survival vs. arrest of mosaic embryos, be more clear about the severity of the mitotic error and its timing (how widely across the embryo, how many chromosomes are affected) and how this contributes to arrest. It is increasingly clear that abnormal mitotic divisions are the main reason why embryos arrest prior to implantation, so the penultimate sentence of the abstract is a bit misleading.

o Please provide additional context about what is known about this process in other species, including the frequency of mitotic errors and the mechanism(s) by which they arise.

- Results

o What happens to micronuclei in subsequent cell divisions?

o When discussing whether embryos were mononucleated after the second division ("the 4-cell stage, 98.4% were mononucleated"), are you implying that the embryos that were multinucleated after the first cell division somehow got corrected? Or are those embryos being excluded in this calculation?

o When comparing mononucleated embryos to those with nuclear defects, you found no difference in rates of miscarriage. But there are many other pregnancy outcomes and selection biases that occur upstream of potential miscarriage that may be relevant. For example, what about rates of preimplantation arrest; rates of blastocyst formation; rates of implantation; embryo morphology; embryo transfer; etc.?

o Trisomy/monosomy rescue are noted as plausible mechanisms to explain the observed patterns of mosaicism. However, you may want to note that uniparental disomy (which is predicted to result from some fraction of such rescue events) is extremely rare in genetic testing data. Thus this mechanism is likely a very small part of the picture.

- Discussion

o Even if there are lower error rates of chromosome mis-segregation in the second and subsequent mitotic divisions, the number of cell divisions occurring increases exponentially. It would be interesting to discuss how those factors (reduced error rate along with greater opportunity for error) relate to one another to produce the observed patterns.

o Please provide additional context about the change in timing of cell division throughout development. Why are later cell divisions faster (e.g., cell size, dilution of maternal gene products,

embryonic genome activation)?

o Please remove the statement that your study “suggests that aneuploid states have been positively selected for during human evolution.” This is extremely speculative and not addressed or supported by your study. While the evolutionary basis of aneuploidy is very interesting, I think it would be better to frame this as an open question and not suggest any particular mode of natural selection.

o As noted previously, I think there is room to better contextualize your study within the broader field, especially given that almost all of the previous literature about aneuploidy and mosaicism in human embryos is based on retrospective analysis of clinical genetic testing data. How do we reconcile your results with that body of work?

- Methods

o Describe how and when the egg-share material was separated into research and treatment sets (e.g., the “clinical decision-making”)

Minor point-by-point comments:

- Introduction

o Recommend mentioning that aneuploidies can also occur in spermatogenesis

(<https://doi.org/10.1038/s41586-020-2347-0>), though contributions from such sperm are rare

o Recommend expanding on the significance of this work in the context of IVF/ART. How will this information contribute to ART?

o Recommend expanding this sentence “The origin of these aneuploid mosaics must be the early mitotic cleavage divisions” (e.g., “Because these aneuploidies affect only a subset of embryonic cells, the origin of these aneuploid mosaics must be the early mitotic cleavage divisions” or something along those lines)

- Results

o May need to define at first use: IVF, 2PN, NEBD, SiR-DNA, IQR

o Can time for pronuclear breakdown be reported in seconds instead of minutes?

o Include counts of embryos in-text when discussing Figure 2a,b and regarding the cohort from 80 clinical time-lapse movies

o Recommend clarifying language of “However, it remains unknown whether the same mechanisms are present in human embryos” (e.g., “However, whether micronuclei similarly contribute to aneuploidies in early human embryonic cells remains unknown” or something along those lines)

o Either change “This confirms” to “this indicates”, or include evidence for why it confirms

o For statements such as “however the rest displayed heterogenous phenotypes” please provide some quantitative measure in the text itself

- Discussion

o The sentence “Such later events would be required in order to explain the wide variation in the number of aneuploid blastomeres per embryo” may require additional explanation. Why would later mitotic errors be expected to impact the variance in rates of mosaicism?

o I also had trouble following the sentence “How pronuclear morphology relates to clinical outcome is unclear, however, it may be an important origin of chromosome segregation errors and mosaic aneuploidy arising in the first human mitotic division.” Please consider breaking into multiple sentences and providing additional explanation.

Reviewer #3 (Remarks to the Author):

Comments for the authors

The manuscript NCOMMS-22-00797-T entitled “The first mitotic division of human embryos is highly error-prone” aims to evaluate when errors in chromosome segregation occur during early

embryogenesis in humans that leads to aneuploid mosaics. It has been previously described in other Mammalian systems that mosaic embryos are consequences of errors during early mitotic divisions in the embryos. However, what happens in human embryos has not been evaluated.

Here the authors made a tremendous effort to visualize for first time chromosome segregation during first and second mitotic division in a subset of living human embryos and extrapolate key events during cellular division to a big set of embryos that are being imaged regularly at IVF clinics with less invasive technology. The first thing that I want to highlight, is the development of an egg sharing research program, where women donate half of their eggs to research, and the other half is used for their IVF treatment. I believe that this program is highly valuable not only for the results generated in this study but also open a huge number of opportunities to improve our knowledge about early processes in human embryos that in another way would be impossible to evaluate.

To visualize by live imaging chromosome segregation, embryos obtained by this egg sharing to research program were stained with Sir-DNA. The authors were able to establish, the timing of key events during cellular division as pro-metaphase, metaphase, and anaphase during the first and second mitotic division in human embryos. More importantly, they observed for the first time that ~30% of the human embryos have lagging chromosomes during the first mitotic division and this percentage is significantly reduced in the second meiotic division. Interestingly, authors find a similar percentage of error in chromosome segregation in the first mitotic division (~30%) in deselected embryos, suggesting that deselected embryos that otherwise will be discarded could be a very good tool to study human early development. Then, the authors correlate the timing of pronuclear breakdown, furrow ingression, and cytokinesis during the first mitotic division between those embryos imaged with Sir-DNA and embryos from the same women imaged regularly at IVF clinics with a less invasive technology (Embryo-scope). Because they observed that the timing of these macro-cytological events was similar, they compare a big cohort of embryos imaged by embryo-scope that were categorized based on if those embryos were transferred or not and if they result in pregnancy or not. Overall, compiling all those embryos there were no differences in the timing of the first mitotic events, however, there are small differences between categories that it would be important to highlight (see my comment below). Finally, the authors tried to correlate chromosome segregation errors with macroscopic features that could be detected by imaging with the embryo scope. Altogether, this study provides very critical information about how the first couple of mitotic division occurs in human embryos, however, I have some concerns about the author's interpretation of some results that I would like to be clarified:

1. The authors used Sir-DNA to track chromosome segregation in human embryos and based only on the pattern of chromosome distribution they assume that the spindle is bipolar or multipolar. Considering only the images that the authors provide in Figures 1 and 2, it is very hard to tell what is going on with the spindle phenotype. In my experience, the pattern of chromosome misalignment is not always a synonym of a multipolar spindle. I would love to see a similar experiment, but this time, stained with Sir-Tub, but I understand the limitation in the access of human embryos. Then, I strongly believe that the authors have to be more cautious at the time to define the abnormality of the embryo categorizing it with spindle phenotype. Formally they do not evaluate the spindle.
2. Similar comment about the spindle phenotype in Figure suppl 2a. Based on chromosome configuration, the authors said that there is evidence of two perpendicular spindles. It is hard to infer that configuration. In panel b, it is easier to see that two spindles are in parallel.
3. In figure 2f, the authors compare the timing of mitotic progression in a big cohort of embryos that they categorized based on if the embryos were transferred or not and lead to pregnancy. However, they do not recall the differences between categories. If the timing was different among categories and speculate about what could be the meaning of that difference.
4. Figure 4. Authors said, "It is well established that lagging chromosomes can lead to the formation of micronuclei (as shown in Fig. 2d) and multipolar chromosome segregation can lead to multinucleated daughter cells." Can the authors provide some references that validate this statement?
5. Following the comments before, the authors said: " We hypothesize that these small nuclei represent micronuclei that form around single lagging chromosomes". I wonder if, in the embryos that they image with Sir-DNA that have lagging chromosomes, they end up forming the small nuclei. That could be an important observation that will validate the way that authors classified the 2-cell embryo

in figure 4. I am not convinced that a single lagging chromosome will form what the authors call small nuclei. Moreover, how do the authors define Large, medium, small... what is the cut-off among categories?

6. In the last paragraph of the discussion, the authors said: "Finally, our finding that the first division is error-prone, combined with the known mosaic nature of human embryos, suggests that aneuploid states have been positively selected for during human evolution." Can the authors elaborate on this? Although it is an interesting idea, I am not sure what they are suggesting with this statement.

Response to reviewers: Manuscript NCOMMS-22-00797-T

Reviewer #1 (Remarks to the Author):

In the manuscript 'The first mitotic division of human embryos is highly error prone', the authors use advanced imaging techniques using a nuclear stain to visualize chromosome segregation errors in the first and second cell divisions in human embryos. These observations are made on embryos generated from donated oocytes and embryos deemed unsuitable for patients' treatment due to delayed or mis-fertilization. The authors also analyze images from clinical time-lapse cases, visually evaluating signs of mitotic errors. Aside from its biological significance, the topic is clinically relevant, considering that early mitotic errors are the source of chromosomal mosaicism in human embryos; there is an ongoing intense debate on how embryos with evidence of mosaicism should be managed in the IVF clinic.

The strong points of the paper are that 1) a method has been set up to visualize live chromosome dynamics in early human embryos, and 2) some of the proposed mechanisms giving rise to mosaicism (previously observed in other organisms) do in fact play a role in human embryos (anaphase lag and tripolar mitosis) and 3) these errors are much more common during the first cell division than during the second cell division.

The manuscript is well written, the references are appropriate, and the figures are well designed and clear.

As soon as the authors start making claims about the incidence of mitotic errors during early divisions in human embryos, they need to tread very carefully. Their sample size of egg-share embryos is admittedly small (understandably, those are very hard to come by), and the population of deselected embryos is not representative of the general embryo population, being highly selected for abnormalities. This is why I consider it a stretch to claim that the first mitotic division is generally 'highly error prone'. In truth, very little can be said from these experiments about the overall proportion of embryos with early mitotic errors. I think the authors might consider softening their language in the text when discussing the incidence/frequency/percentage of early mitotic errors in the general embryo population (for example, the first sentence of the Discussion, but also recurrently in the text). On the other hand, the experiments certainly suggest that the first mitotic division is significantly more error prone relative to the second mitotic division, which is a point that should be emphasized.

1.1 Our conclusion that human embryos are highly error prone is based on comparison to work showing that the frequency of anaphase lagging chromosomes (caused by merotelic attachments) in non-transformed human cells is only 2-6% (Bakhoun et al, 2014, Current Biology). (line 118-122) Importantly, the frequency of lagging chromosomes is similar in deselected and egg sharer (gold standard) embryos. The mitotic timing of these sets of embryos is also similar providing further evidence that deselected embryos are more representative of the general population than previously thought. These data together allow us to conclude that error-generating phenotypes are frequent in the first mitotic division and state that this is indicative of an elevated incidence of chromosome mis-segregation. Nevertheless, we agree on caution here and have softened the language throughout the paper, made reference to key literature that shows aneuploidy in early human embryos, and made sure to emphasize the difference with the second division.

Major Points:

- I was not able to access any of the original Movies. There are plenty of representative movie stills in the Figures, but I would like to see actual clips of the time lapse imaging with

SiR-DNA stain, as well as the clinical movies/Embryoscope, showing some of the mitotic events discussed in the manuscript. Such visuals would definitely elevate the paper.

1.2 We apologise for this and hope that now movies are viewable via OMERO using the following link:

https://warwick.ac.uk/fac/sci/med/research/biomedical/facilities/camdu/public_data/

Please use the login and password 'public' to access the dataset for this paper.

- Using SiR-DNA the authors can visualize processes which they interpret as mitotic chromosome segregation errors, but they never actually confirm such events by molecular means. For example, if an anaphase lag is apparently observed at the first division, could the authors do chromosomal testing of the two daughter cells (or even later, at the four-cell stage) to confirm that indeed there is a chromosome number error in one or more cells? This can be done by common PGT-A platforms, as I don't see how SiR-DNA treatment could interfere with PGT-A chemistries. I think such a validation would considerably increase the confidence in their system consequently in the conclusions the manuscript makes.

1.3 This is an excellent suggestion. We are currently establishing a PGT-A workflow to analyse the karyotypes of the daughters of the first division following live cell imaging. However, this will require an entire new data set and patient cohort, as we have had to seek additional ethical approvals to allow patient consenting for genetic testing. This experiment is thus well beyond the scope of the current study. We have, however, made clear in the text that there is evidence in the literature that chromosome loss and gain do occur in the early divisions, and this has been confirmed by sequencing in these studies. We have edited our text to emphasise that this study provides fresh insight into the cell biological events that underlie chromosome mis-segregation during the first and second mitotic divisions.

- Regarding the clinical samples visualized by Embryoscope, the authors hypothesize that the small nuclei observed in some samples are micronuclei that form around lagging chromosomes. Again, this could be substantiated by molecular methods and showing that indeed daughter cells contain chromosome copy number errors. By the way, confirming a correlation between small nuclei and aneuploidy would be quite impactful. Many IVF clinics use time-lapse imaging such as Embryoscope, and this could be a way to quickly and easily predict aneuploidy in embryos.

1.4 Please see comment above regarding PGT-A and patient consent. Furthermore, these embryos were used for patient fertility treatment and were transferred to the patient, therefore the EmbryoScope movies are the only data we can collect for this specific cohort. We have made it clear in the text that previous studies have confirmed aneuploidy in human embryos containing micronuclei (Line 215-217).

- Figure 4a: assessment of nuclei seems highly subjective, at least judging from the images shown. If the authors have other sample images of better quality, they should consider exchanging them. Was there any software or standardized method used to determine if a nucleus was 'large', 'medium', or 'small'? It says in the methods that a trained embryologist evaluated the nuclei. But my concern is that some of those dots might not even be micronuclei. It would be more convincing if the authors could take some of those embryos with visual evidence of micronuclei by Embryoscope and do a nuclear stain, showing that the small nuclei and/or 'micronuclei' do indeed contain chromosomal material.

1.5 We thank the reviewer for this comment and have quantitatively reanalysed the size distribution of nuclei in Figure 6 (Lines 209-228). To do this we created a FIJI macro that assimilates image stacks from EmbryoScope movies to better visualize nuclei and then accurately measured nuclear diameter. This re-analysis showed that a few more embryos contained abnormal sized nuclei than previously thought,

reflecting our improved analysis workflow. The distribution of nuclear sizes is shown in Figure 6c and reveals two broad populations: the first reflecting “normal” sized nuclei and a second that reflects various sized smaller nuclei. The smallest of these (<10µm) represent micronuclei, and we have included example movies to better illustrate phenotypes. We note that the variation in sizes of the micronuclei population is consistent with previous studies (doi: 10.1016/0165-1218(85)90067-9.).

Please also see above comment (1.4) regarding use of patient clinical embryos for research purposes which prevents us staining these embryos with a nuclear stain at the 2-cell stage. However, our chromosome imaging of deselected embryos shows similar sized micronuclei forming that contain DNA (as stained with SiR-DNA and H2B mCherry) (Figure 2D and Figure 3).

- The findings might not be entirely representative of clinical-use embryos (let alone embryos from natural conceptions!), since the experimental embryos were exposed to several manipulations (zona pellucida removal by chemical treatment, addition of SiR-DNA, transport to imaging site, and special imaging method). I am also still not convinced that SiR-DNA treatment does not influence the rate/type of mitotic error. The authors mention that overall, the developmental timeframes are similar between treated and untreated embryos, but I don't think stage progression is a sufficiently informative metric regarding mitotic errors and whether their protocol can induce them. For example, I cannot agree with the statement: 'This confirms that chromosome imaging does not perturb the first embryonic mitosis...'. SiR-DNA interaction with DNA could possibly interfere with the kinetochore-microtubule complex, thereby inducing mitotic mistakes. These concerns and caveats should be dispelled with further experiments or at the very least carefully explored in the discussion.

1.6 To control for use of SiR-DNA to visualize chromosomes, we performed H2B-mCherry mRNA injection in a very small cohort of deselected embryos and imaged these on a spinning disk confocal microscope which is gentler on the sample (Lines 115-162). This method has been used extensively in the literature, most notably to visualise chromosomes during human meiosis (Holubcová et al, 2015, Science). This reduces the manipulations required prior to imaging as the zona pellucida remains intact. We were able to see a similar, if not higher, frequency of aberrant mitosis using this visualisation and imaging method as the resolution is far better and defects are more easily visible.

These new data are presented in a new Figure 3.

We also note that SiR-DNA has been reported to have no influence on the duration of mitosis or chromosome error rate when compared to H2B-mCherry chromatin visualisation in human somatic cells (Lukinavičius et al, 2015). One study found that SiR-DNA can induce a DNA damage response at high concentrations, and this results in G2 arrest and subsequent mitotic block (Sen et al, 2018, Scientific Reports). However, all the embryos in our dataset progressed through mitosis, suggesting that our SiR-DNA treatment is not perturbing G2.

We have expanded the introduction to expand on the existing literature that describes the high occurrence of aneuploidies in human preimplantation embryos, indirectly observed via preimplantation genetic testing (lines 58-70).

- What is happening between the first and second division that increases the fidelity of chromosome segregation? Why is the first division much more error prone than the second? They mention the ratio of cell size to spindle in the second-to-last paragraph of the discussion, but the authors should elaborate and/or offer additional explanations (since in my mind is the major biological finding of the study).

1.7 Identifying the underlying mechanism for increased errors in the first mitosis would prove very interesting. There are several possibilities which we now include in the discussion: 1) The first mitotic division is unique in that it is the first time the parental haploid sets are segregated together. Zygote genome unification, in terms of nucleoli clustering at the pronuclei interface, is error prone in bovine embryos, and this is associated with subsequent chromosome errors during the first mitotic division (Cavazza *et al.*, 2021 <https://doi.org/10.1016/j.cell.2021.04.013>). 2) Dual mitotic spindles are thought to separately segregate maternal and paternal genomes during the first mitotic division in bovine (non-rodent) embryos, in which centrioles are inherited from the sperm (Reichmann *et al.*, 2018 DOI: [10.1126/science.aar7462](https://doi.org/10.1126/science.aar7462); Schneider *et al.*, 2021 doi: [10.1083/jcb.202010106](https://doi.org/10.1083/jcb.202010106)). Failure to combine the dual spindle causes major chromosome errors. 3) The role of cell size is debated in the literature. Large cell size is linked to decreased SAC strength in mouse oocytes (Kyogoku and Kitajima, 2017 doi: [10.1016/j.devcel.2017.04.009](https://doi.org/10.1016/j.devcel.2017.04.009)), but not early embryos (Vázquez-Diez, Paim and FitzHarris, 2019 doi: [10.1016/j.cub.2018.12.042](https://doi.org/10.1016/j.cub.2018.12.042)). However, SAC strength in cells of developing *C. elegans* embryos increases with decreasing size (Galli and Morgan, 2016 DOI: [10.1016/j.devcel.2016.01.003](https://doi.org/10.1016/j.devcel.2016.01.003)). We have expanded the discussion to include these points, but we believe that testing these hypotheses is beyond the scope of this study. This can be found in lines 272-294.

Minor points:

- Figure 1c: Why was PNBD not observed in 6 out of 13 embryos for research? And yet, it was observed in all 16 embryos used for treatment? In view of those differences, can the following really be claimed: 'suggesting that the fluorescent imaging methodology did not impair embryo progression.' On the contrary, there seems to be a clear difference between the embryos treated with SiR-DNA and the others. Please clarify.

1.8 This is simply due to not getting the research embryos back to the imaging facility before PNBD occurred. Treatment embryos went into the EmbryoScope immediately following identification of 2PN fertilisation after IVF or immediately after sperm injection for ICSI, so all stages were filmed. This has been clarified in the figure legend.

- Figure 1: embryo number 3 of patient 3247 of Figure 1c has an asterisk on it and refers to the embryo shown in 1b. However, I thought that egg-share-to-research embryos were free from the zona pellucida, and yet it is still present in 1b. See also the text on this point: 'In parallel, patient treatment embryos were monitored in the clinic via time-lapse imaging with Hoffman contrast (EmbryoScope) to inform embryo selection (Fig. 1b).

1.9 This is an error which has been corrected, it was meant to refer to 1a.

- Figure 1c: Indicate which embryo(s) are depicted in 1e.

1.10 This has now been added to the figure legend.

- Figure 1c: What happened to the embryo 4 of patient 3247 in Figure 1c? What about embryo 2 of patient 3371 in Figure 1c? Are those instances where the zygote immediately divides? Please clarify.

1.11 Please see explanation above for point 1.8. They started dividing before we could get them to the research lab.

- Fig 2: How were the embryos 'unsuitable for transfer' used? Were they frozen at the zygote stage, donated, then thawed and used for the experiment? Or did the patient consent to donate such embryos ahead of time? I suspect it's the second one, but then, were the zonae removed for this group of embryos as well? Please clarify these points in the manuscript.

1.12 These clinical treatment embryos become correctly fertilised with the view to either freeze at blastocyst stage or transfer to the patient. However, the embryologist decided they were not high enough grade to be used, so these embryos were discarded at the end of EmbryoScope observation (day 6). The patients only consented to use of their Embryoscope movies in research. These actual embryos were never handled by research staff, therefore have an intact zona pellucida. We have clarified this in the methods. Please be aware this data has moved to figure 4.

- Figure 2 c: Why are 1PN and 2PN grouped together?

1.13 3PN embryos can be the result of fertilisation by two sperm. This is important as the sperm donates the centrosomes by which the mitotic spindle is built, and a 3PN embryo could contain supernumerary centrosomes (from two sperm), resulting in a higher probability of forming a multipolar spindle at the first mitosis. 1PN embryos have potential to be genetically normal as they can also form due to pronuclear fusion and would likely have one centrosome (from one sperm). Therefore we have separated 1PN and 2PN embryos from 3PN embryos for analysis of multipolar divisions, to be sure that multipolar divisions are not only a feature of embryos with supernumerary centrosomes.

- Figure 3 d and e: What does e) tell us that cannot already be deduced from d)?

1.14 We wanted to specifically highlight the overall reduction of lagging chromosomes in bipolar divisions between the first and second mitosis, as this is a major finding. Please note that this data has been moved to figure 5.

- Figure 4c: 4 embryos have gone missing 50+18 in the left column but only 64 in the right column. What happened to those 4 cases?

1.15 We have now reanalysed this whole dataset, but this was due to the embryologist determining nuclei as unclear and therefore discarded them from the data. These data are now in Figure 6.

- Figure 4e: My understanding is that nondisjunction and anaphase lag are two different mechanisms, one leading to reciprocal aneuploidies in daughter cells, the other leading to a loss of chromosomes in one daughter cells (see doi: 10.1093/humupd/dmu016). I don't see how the authors are combining both mechanisms in their model (Fig 4e). Their SiR-DNA imaging system does not allow the authors to visualize nondisjunction (the extra chromosome going to one daughter cells cannot be observed at this resolution). I don't think any claims can be made regarding nondisjunction based on the data presented here.

1.16 We believe this is a wider issue due to inconsistent use of language, the term nondisjunction refers to when both the replicated chromatids fail to segregate and end up in one daughter cell. 'True nondisjunction' can refer to the situation when sister chromatids fail to separate at anaphase, and we agree that this could not be determined from our movies. Anaphase lag is a cell biological phenomenon where chromatids are visibly "lagging" between separating chromosome clusters. This is

generally due to merotelic attachment of one or both kinetochores. The result of this can be nondisjunction (both sisters into one daughter) or it can lead to the formation of micronuclei (leading to chromothripsis which causes structural aneuploidies or even loss of the chromosome completely). We have updated our model (figure 7) and discussion to clarify differences in these causes of aneuploidy (lines 242-261).

- Supplementary Fig 2 a): It's very hard to distinguish anything in the siR-DNA panels (including the high magnifications). Are there any better ones that can be used?

1.17 We cannot improve the resolution using the widefield microscope. In future we aim to image using confocal spinning disc microscopy (as in figure 3) to give better quality movies. The new figure 3 documents similar events seen in widefield but with improved resolution.

- There are two papers that the authors should consider incorporating in their manuscript as they are highly relevant to the discussion of mosaic embryo management in the IVF clinic: doi:10.1093/humrep/dez309 and doi: 10.1016/j.fertnstert.2020.11.041

1.18 We thank the reviewer for these references and have included these in the revised text.

Reviewer #2 (Remarks to the Author):

In their submitted manuscript, Ford et al. apply a novel approach for visualizing chromosome dynamics during the first two cell divisions of the human embryo, quantifying various aspects of mitotic progression. Overall, this work is interesting and important for the fundamental understanding of human development, which in turn provides a foundation for fertility therapies, etc. The comments that follow largely focus on providing additional context and clarifying the novel aspects of this work, along with some statistical recommendations that do not require additional experimentation.

Major comments:

- This work is clearly novel in presenting a live-embryo chromosome imaging methodology for humans and related to the phenomenon of mosaicism that is of longstanding interest and debate in the field of human IVF. However, I think the novel aspects of this work in relation to past work can be better emphasized and contextualized, perhaps by expanding the Introduction and Discussion. For example, the sentence "Live-embryo chromosome imaging allows us to determine the timing of key chromosome related events that are invisible in clinical image sequences" in the Results implies aspects of mitotic timing that were previously unknown. This could be stated earlier, in the Introduction, and expanded upon to clarify what exactly was known and unknown (in humans; in other organisms, etc.).

2.1 We thank the reviewer for this really positive comment and agree that we can do more to emphasise the novel aspects of our work. We have done this throughout the text.

- Related to the previous point, can you please clarify for a non-specialist why observation of the chromosomes is necessary for estimating the timing of mitotic events (in contrast to conventional time lapse imaging)?

2.2 We have edited the text to be more accessible to readers of all backgrounds (line 71-80). Chromosomes are not visible using standard time-lapse microscopic

techniques applied in IVF units. Some aspects, such as the time to forming a metaphase plate, and in particular anaphase onset and errors, cannot be observed without visualising chromosomes.

- Many of the Results in the study hinge on the observation that abnormally fertilized embryos available to research had similar developmental properties and could be used to draw conclusions about normally fertilized embryos. The evidence provided was that statistical tests comparing the two groups were not significant, with p-values provided. However, it was unclear to me whether in this small sample, you have enough statistical power that a lack of significant differences is meaningful. One way to address this would be to include confidence intervals along with p-values, so that we know how large of a difference we can rule out. Another way to address this would be to more honestly discuss the limitation in sample size and the fact that abnormally fertilized embryos may not be a perfect proxy for fertilized embryos and how this may impact interpretation of your results.

2.3 We are explicit with regard to the numbers of embryos in each group and have emphasised the limitation of sample size, which is inevitable, given the nature of the material that we are using. Furthermore, we have reanalysed all embryoscope movies using a new ImageJ macro (see reviewer 1, point 1.5) as it improves accuracy. The difference in the median times for research embryos vs. all clinical embryos is only 15 mins (this is only 1 to 1.5 time frames in our live imaging). This is within the population variation and the confidence intervals are now provided along with p-values that are corrected for comparison of multiple groups (Kruskal Wallis and post-hoc Dunn's test). With our sample size we do not have strong evidence for rejecting the null hypothesis (even at 5% level). Nevertheless, we have added a sentence to point out that such small time differences could be relevant although much larger sample sizes and higher temporal resolution live imaging would be needed (line 178). In the past there has been criticism of research using deselected embryos from IVF treatment because they were assumed to be substantially different to 'normal' clinical embryos, but this had not been proven. Our study provides evidence that deselected embryos are similar to 'normal' clinical embryos, which is an important point for those working on such material and for the interpretation of studies in this field. We have rephrased some of the text to address this (Lines 170-182).

- Related to the point above similar logic was used to argue that the imaging and culture conditions did not impact mitotic fidelity, but this should also be moderated as suggested above.

2.4 We believe the significant reduction in segregation errors seen during mitosis 2 addresses this point as it demonstrates that errors observed in the first division are highly unlikely to be a consequence of imaging and culturing conditions, as these embryos undergoing mitosis 2 have been cultured and imaged for ~30 hours. Furthermore, nuclear variants were found in ~25% of blastomeres from a cohort of 2-cell clinical embryos which gave rise to pregnancies and live birth. We hypothesise that these variants originate from an error prone mitosis 1 leading to formation of micronuclei and/or multiple nuclei in these embryos which have not been manipulated, as observed in our chromosome imaging movies. The frequency of micronuclei (<10µm) in clinical embryos (~12%, line 208-226, Figure 6d), is consistent with those seen forming in research embryos during chromosome imaging (~14%, line 126-130, Figure 2b).

- Please provide an explanation for why only the first two cell divisions were tracked. Is this a technical limitation (e.g., degradation of the dye used for imaging chromosomes)? Or a practical limitation (tracking more cells makes data collection too challenging)? What do you speculate you would observe if you tracked additional divisions, especially after embryonic

genome activation?

2.5 This was a practical limitation in that once the embryo divides into 4 cells it is very difficult to track daughter lineages, often in different planes, especially in these embryos where the zona pellucida is removed and daughter cells may move apart. In the future we aim to image further divisions, but this requires substantial optimisation. We would speculate that cells with anomalies spotted earlier would fall behind in subsequent cell divisions, as this is what is observed clinically.

Major point-by-point comments:

- Introduction

o In the discussion of survival vs. arrest of mosaic embryos, be more clear about the severity of the mitotic error and its timing (how widely across the embryo, how many chromosomes are affected) and how this contributes to arrest. It is increasingly clear that abnormal mitotic divisions are the main reason why embryos arrest prior to implantation, so the penultimate sentence of the abstract is a bit misleading.

2.6 We have now provided more emphasis on the heterogeneity of outcomes in the text (line 46-54, 58-70). Interestingly, our research is focused on the first cleavage division where mitotic errors would potentially affect the entire embryo, yet these can remain compatible with healthy euploid birth. We have not studied arresting embryos specifically in this study (others have, e.g. McCoy et al, 2022, bioRxiv <https://doi.org/10.1101/2022.07.03.498614>) but from the literature and clinical practice we consider that in addition to chromosomal errors, there may be other reasons for embryo arrest, such as metabolic insufficiency, membrane deficiencies, inadequate cellular volume, problems with cell orientation, polarity, communication and intercellular junctions, dissociation of cytokinesis and karyokinesis, fragmentation and loss of cellular mass, etc.

o Please provide additional context about what is known about this process in other species, including the frequency of mitotic errors and the mechanism(s) by which they arise.

2.7 We have provided additional context from studies in other species in the introduction and discussion (lines 61-68, 282-294).

- Results

o What happens to micronuclei in subsequent cell divisions?

2.8 We do not have the chromosome imaging data needed to answer this question currently, but our embryoscope nucleation data at the 4-cell stage (figure 6b) suggests that chromosomes in micronuclei could segregate with the rest of the blastomere chromosomes and not be passively inherited. Humans differ from mice in this respect (doi.org/10.1073/pnas.1517628112). Our image in Figure 3c also shows that micronuclei can produce chromosomes that can integrate with the main metaphase plate.

o When discussing whether embryos were mononucleated after the second division ("the 4-cell stage, 98.4% were mononucleated"), are you implying that the embryos that were multinucleated after the first cell division somehow got corrected? Or are those embryos being excluded in this calculation?

2.9 Please see above comment. The data implies some form of 'correction'. Clinical data show that the micronuclei reintegrate to form a single nucleus following the second division (unlike in mouse).

o When comparing mononucleated embryos to those with nuclear defects, you found no difference in rates of miscarriage. But there are many other pregnancy outcomes and selection biases that occur upstream of potential miscarriage that may be relevant. For example, what about rates of preimplantation arrest; rates of blastocyst formation; rates of implantation; embryo morphology; embryo transfer; etc.?

2.10 All embryos in this specific dataset formed blastocysts which were transferred to the patient and gave rise to 7 week fetal heartbeat, so they thus proved their quality in development to at least this stage. We agree that analysis of further clinical embryology endpoints would be interesting, but as a study had been previously published we did not repeat this analysis ourselves. <https://doi.org/10.1093/humrep/deh003>

o Trisomy/monosomy rescue are noted as plausible mechanisms to explain the observed patterns of mosaicism. However, you may want to note that uniparental disomy (which is predicted to result from some fraction of such rescue events) is extremely rare in genetic testing data. Thus this mechanism is likely a very small part of the picture.

2.11 We have clarified the rarity of uniparental disomy in the text (line 250)

- Discussion

o Even if there are lower error rates of chromosome mis-segregation in the second and subsequent mitotic divisions, the number of cell divisions occurring increases exponentially. It would be interesting to discuss how those factors (reduced error rate along with greater opportunity for error) relate to one another to produce the observed patterns.

2.12 We have expanded on this in the discussion (lines 262-271) but do not have data to contribute beyond the 2nd cleavage division.

o Please provide additional context about the change in timing of cell division throughout development. Why are later cell divisions faster (e.g., cell size, dilution of maternal gene products, embryonic genome activation)?

2.13 We feel this is beyond the scope of this paper as we have only examined the first two cleavage divisions, but we have included substantially more discussion on why the first embryonic division is unique (lines 272-394)

o Please remove the statement that your study "suggests that aneuploid states have been positively selected for during human evolution." This is extremely speculative and not addressed or supported by your study. While the evolutionary basis of aneuploidy is very interesting, I think it would be better to frame this as an open question and not suggest any particular mode of natural selection.

2.14 We have removed this section.

o As noted previously, I think there is room to better contextualize your study within the broader field, especially given that almost all of the previous literature about aneuploidy and mosaicism in human embryos is based on retrospective analysis of clinical genetic testing data. How do we reconcile your results with that body of work?

2.15. As the referee notes, most literature is based upon retrospective analysis of PGT-A in human embryos. This methodology is suboptimal because it analyses (usually) a fraction of the trophectoderm at the blastocyst stage. Earlier embryos were usually biopsied by removing one or two cells and analysed by earlier genetic tests which were less reliable and comprehensive in their analysis. Clinically, no

substantial benefit of PGT-A for pregnancy has been demonstrated by large clinical trials, though this remains highly controversial and at odds with scientific expectations, perhaps highlighting confounding influences. Separate analysis of all cells in an embryo at different stages is needed to understand the evolution of mitotic aneuploidy, but this is technically and ethically challenging. Perhaps through the use of egg share to research embryos, such analysis can be considered, but for now it has not been undertaken.

We have expanded the introduction to include more background and we have aimed to contextualise our study within the broader field (lines 46-57), but due to our concerns about the suitability of PGT-A as a reliable indicator of whole embryo chromosome content, and its apparent lack of clinical efficacy, PGT-A has not been used as a major source of our reference material for this study. We prefer to approach this from the basic science of mitosis in somatic cells (for comparison), and clinical expertise in embryos unselected for genetic content.

- Methods

o Describe how and when the egg-share material was separated into research and treatment sets (e.g., the "clinical decision-making")

2.16 Substantial additional detail of clinical methods and decision-making regarding egg sharing and fertilisation checks has been included (lines 323-339)

Minor point-by-point comments:

- Introduction

o Recommend mentioning that aneuploidies can also occur in spermatogenesis (<https://doi.org/10.1038/s41586-020-2347-0>), though contributions from such sperm are rare

o Recommend expanding on the significance of this work in the context of IVF/ART. How will this information contribute to ART?

o Recommend expanding this sentence "The origin of these aneuploid mosaics must be the early mitotic cleavage divisions" (e.g., "Because these aneuploidies affect only a subset of embryonic cells, the origin of these aneuploid mosaics must be the early mitotic cleavage divisions" or something along those lines)

2.17 We thank the reviewer for these comments and have included them in the introduction accordingly.

- Results

o May need to define at first use: IVF, 2PN, NEBD, SiR-DNA, IQR

o Can time for pronuclear breakdown be reported in seconds instead of minutes? –

We do not have the time resolution to do this (chromosome imaging time resolution is 15 mins, embryoscope resolution is reported as a fraction of the hour).

Include counts of embryos in-text when discussing Figure 2a,b and regarding the cohort from 80 clinical time-lapse movies

o Recommend clarifying language of "However, it remains unknown whether the same mechanisms are present in human embryos" (e.g., "However, whether micronuclei similarly contribute to aneuploidies in early human embryonic cells remains unknown" or something along those lines)

o Either change "This confirms" to "this indicates", or include evidence for why it confirms

o For statements such as “however the rest displayed heterogenous phenotypes” please provide some quantitative measure in the text itself

2.18 We have edited the text accordingly to incorporate these points

- Discussion

o The sentence “Such later events would be required in order to explain the wide variation in the number of aneuploid blastomeres per embryo” may require additional explanation. Why would later mitotic errors be expected to impact the variance in rates of mosaicism?

2.19 As mis-segregations arising later in embryo development would give rise to a ‘smaller’ lineage of cells compared to ones arising in the first mitotic division. We have clarified this point in lines 266-267

o I also had trouble following the sentence “How pronuclear morphology relates to clinical outcome is unclear, however, it may be an important origin of chromosome segregation errors and mosaic aneuploidy arising in the first human mitotic division.” Please consider breaking into multiple sentences and providing additional explanation.

2.19 We have edited the text accordingly (lines 282-288)

Reviewer #3 (Remarks to the Author):

Comments for the authors

The manuscript NCOMMS-22-00797-T entitled “The first mitotic division of human embryos is highly error-prone” aims to evaluate when errors in chromosome segregation occur during early embryogenesis in humans that leads to aneuploid mosaics. It has been previously described in other Mammalian systems that mosaic embryos are consequences of errors during early mitotic divisions in the embryos. However, what happens in human embryos has not been evaluated.

Here the authors made a tremendous effort to visualize for first time chromosome segregation during first and second mitotic division in a subset of living human embryos and extrapolate key events during cellular division to a big set of embryos that are being imaged regularly at IVF clinics with less invasive technology. The first thing that I want to highlight, is the development of an egg sharing research program, where women donate half of their eggs to research, and the other half is used for their IVF treatment. I believe that this program is highly valuable not only for the results generated in this study but also open a huge number of opportunities to improve our knowledge about early processes in human embryos that in another way would be impossible to evaluate.

To visualize by live imaging chromosome segregation, embryos obtained by this egg sharing to research program were stained with Sir-DNA. The authors were able to establish, the timing of key events during cellular division as pro-metaphase, metaphase, and anaphase during the first and second mitotic division in human embryos. More importantly, they observed for the first time that ~30% of the human embryos have lagging chromosomes during the first mitotic division and this percentage is significantly reduced in the second meiotic division. Interestingly, authors find a similar percentage of error in chromosome segregation in the first mitotic division (~30%) in deselected embryos, suggesting that deselected embryos that otherwise will be discarded could be a very good tool to study human early development. Then, the authors correlate the timing of pronuclear breakdown, furrow ingression, and cytokinesis during the first mitotic division between those embryos imaged with Sir-DNA

and embryos from the same women imaged regularly at IVF clinics with a less invasive technology (Embryo-scope). Because they observed that the timing of these macro-cytological events was similar, they compare a big cohort of embryos imaged by embryo-

scope that were categorized based on if those embryos were transferred or not and if they result in pregnancy or not. Overall, compiling all those embryos there were no differences in the timing of the first mitotic events, however, there are small differences between categories that it would be important to highlight (see my comment below). Finally, the authors tried to correlate chromosome segregation errors with macroscopic features that could be detected by imaging with the embryo scope. Altogether, this study provides very critical information about how the first couple of mitotic division occurs in human embryos, however, I have some concerns about the author's interpretation of some results that I would like to be clarified:

3.0. We thank the referee for their positive evaluation and acknowledgement of our work and the wider potential of our egg sharing programme. We have noted and highlighted in the text the small differences between categories and our interpretation of these (lines 180-182).

1. The authors used Sir-DNA to track chromosome segregation in human embryos and based only on the pattern of chromosome distribution they assume that the spindle is bipolar or multipolar. Considering only the images that the authors provide in Figures 1 and 2, it is very hard to tell what is going on with the spindle phenotype. In my experience, the pattern of chromosome misalignment is not always a synonym of a multipolar spindle. I would love to see a similar experiment, but this time, stained with Sir-Tub, but I understand the limitation in the access of human embryos. Then, I strongly believe that the authors have to be more cautious at the time to define the abnormality of the embryo categorizing it with spindle phenotype. Formally they do not evaluate the spindle.

3.1 We thank the reviewer for this comment and agree. Our interpretation of spindle as bipolar or multipolar is mainly based upon the separation of chromosomes into two or more groups at anaphase, which is a reliable marker. However, before anaphase, determination of spindle anomalies based upon chromosome configurations is far less reliable. We did not use SiR-Tubulin as it is thought to stabilise microtubules (due to it being Taxol derived), but instead, to address this query, we fixed some embryos during prometaphase and metaphase of the first mitotic division and stained with kinetochore and tubulin antibodies (Figure 2e). We observed a mixture of bipolar and multipolar spindles which seems consistent with our live imaging observations (line 150-154).

2. Similar comment about the spindle phenotype in Figure suppl 2a. Based on chromosome configuration, the authors said that there is evidence of two perpendicular spindles. It is hard to infer that configuration. In panel b, it is easier to see that two spindles are in parallel.

3.2 We have now supplied movie files and hope that this improves visualisation of this phenotype. Movies are viewable via the following link: https://warwick.ac.uk/fac/sci/med/research/biomedical/facilities/camdu/public_data/ Please use the login and password 'public' to access the dataset for this paper.

We agree that our study is limited due to not being able to observe the spindle during live imaging, but we hope that our new fixed embryo immunofluorescence images can support these conclusions (figure 2e). We have toned down the language for this section.

3. In figure 2f, the authors compare the timing of mitotic progression in a big cohort of embryos that they categorized based on if the embryos were transferred or not and lead to pregnancy. However, they do not recall the differences between categories. If the timing was

different among categories and speculate about what could be the meaning of that difference.

3.3 See comment 2.3 to reviewer 2. We have now highlighted the small differences in timings between the different groups of embryos (now in Figure 4), however, these differences are relatively minor and the overall impression is that the groups are substantially similar.

4. Figure 4. Authors said, "It is well established that lagging chromosomes can lead to the formation of micronuclei (as shown in Fig. 2d) and multipolar chromosome segregation can lead to multinucleated daughter cells." Can the authors provide some references that validate this statement?

We have added new references:

doi.org/10.1530/REP-17-0569

doi.org/10.1093/mutage/geq052

5. Following the comments before, the authors said: "We hypothesize that these small nuclei represent micronuclei that form around single lagging chromosomes". I wonder if, in the embryos that they image with Sir-DNA that have lagging chromosomes, they end up forming the small nuclei. That could be an important observation that will validate the way that authors classified the 2-cell embryo in figure 4. I am not convinced that a single lagging chromosome will form what the authors call small nuclei. Moreover, how do the authors define Large, medium, small... what is the cut-off among categories?

3.4 We provided an example of a lagging chromosome directly forming a micronucleus in figure 2d, plus further new examples in a new figure 3, visualised using H2b-mCherry and spinning disk confocal microscopy.

We have now quantitatively reanalysed the size distribution of nuclei in figure 6 (line 209-228). To do this we created a FIJI macro that assimilates image stacks from EmbryoScope movies to better visualise nuclei and then accurately measured nuclear diameter. This re-analysis showed that a few more embryos contained abnormal sized nuclei than previously thought, reflecting our improved analysis workflow. The distribution of nuclear sizes is shown in Figure 6c and reveals two broad populations: the first reflecting "normal" sized nuclei and a second that reflects various sized smaller nuclei. We think that that smallest of these (<10µm) could represent micronuclei. We have included example movies to better illustrate this phenotype. We note that the variation in sizes of the micronuclei population is consistent with previous studies (doi: 10.1016/0165-1218(85)90067-9.). The text can be found in lines 203-299.

6. In the last paragraph of the discussion, the authors said: "Finally, our finding that the first division is error-prone, combined with the known mosaic nature of human embryos, suggests that aneuploid states have been positively selected for during human evolution." Can the authors elaborate on this? Although it is an interesting idea, I am not sure what they are suggesting with this statement.

3.6 We have now removed this section from the text, as suggested by reviewer 2.

REVIEWERS' COMMENTS

Reviewer #1 (Remarks to the Author):

I was still unable to access the original Movies. When I click on the link the message says 'We're sorry, but the page you requested could not be found'. This should be fixed before publication.

However, all other points I raised have been addressed adequately.

Reviewer #2 (Remarks to the Author):

I appreciate the authors' consideration of my comments, as well as those of the other reviewers. I am broadly satisfied with these revisions. The positioning of their method and findings in the context of the field (clinical reproductive genetics as well as developmental biology / cytogenetics) is useful and increases the impact of the study. The quantitative analysis of nuclei size (Figure 6) and the discussion of the first division's propensity for error further support their novel conclusions. This is an important contribution to the literature on the fidelity of early embryonic mitosis.

Note that I still could not access the videos at the link the authors provided, but I trust these will be made available and evaluated by the other reviewers (who requested them) if necessary.

Reviewer #3 (Remarks to the Author):

The authors answered all my concerns and I consider the inclusion of the evaluation of the spindle in embryos by immunofluorescence, the live imaging of embryos expressing H2B, and the new analysis of the pronuclei size showing the distribution of sizes, to make more powerful the conclusions of this study. Moreover, I agree with the discussion of the different hypotheses about why the first mitosis is more error-prone. It is useful to guide the following studies needed to understand the mechanism behind it.